# Global Convergence and Pareto Front Exploration in Deep-Neural Actor-Critic Multi-Objective Reinforcement Learning

## Abstract

Multi-objective reinforcement learning (MORL) has gained considerable traction in recent years, with applications across diverse domains. However, its theoretical foundations remain underdeveloped, especially for widely used but largely heuristic deep neural network (DNN)-based actor–critic methods. This motivates us to study MORL from a theoretical perspective and to develop DNN-based actor–critic approaches that (i) provide global convergence guarantees to Pareto-optimal policies and (ii) enable systematic exploration of the entire Pareto front (PF). To achieve systematic PF exploration, we first scalarize the original vector-valued MORL problem using the weighted Chebyshev (WC) technique and leveraging the one-to-one correspondence between the PF and WC scalarizations. We then address the non-smoothness introduced by WC in the scalarized problem via a parameterized log-sum-exp softmax approximation, which allows us to design a deep neural actor–critic method for solving the smoothed WC-scalarized MORL problem with a global convergence rate of $\mathcal{O}(1/T)$, where $T$ denotes the total number of iterations. To the best of our knowledge, this is the first work to establish theoretical guarantees for both global convergence and systematic Pareto front exploration in deep neural actor–critic MORL. Finally, extensive numerical experiments and ablation studies on recommendation system training and robotic simulation further validate the effectiveness of our method, especially its capability in Pareto exploration.

## 1 Introduction

**1) Background and Motivations.** Although traditional reinforcement learning (RL) has made remarkable strides over the past few decades (Kaelbling et al., 1996; Sutton et al., 1998; Arulkumaran et al., 2017), as the machine learning paradigms become increasingly complex, it struggles to model some real-world scenarios that involve multiple underlying objectives. Take reinforcement learning with human feedback (RLHF) as an example: multiple human-aligned metrics, such as *helpfulness*, *verbosity*, and *toxicity*, may conflict with each other (Ouyang et al., 2022; Wang et al., 2023; Chakraborty et al., 2024), making it insufficient to only adopt a *single* reward signal to represent them. Consequently, this has motivated the research on the multi-objective reinforcement learning (MORL) (Gábor et al., 1998; Van Moffaert & Nowé, 2014; Yang et al., 2019), which seeks to maximize multiple reward functions. In MORL, due to the potentially conflicting nature of objectives, it is generally impossible to find a single policy to maximize them simultaneously. Therefore, one typically aims to find an optimal policy in the Pareto sense, meaning that the performance of any single objective cannot be further improved without compromising other objectives.

As a subfield of reinforcement learning (RL), MORL problems can potentially be tackled by various fundamental RL approaches. Among them, the actor-critic approach (Sutton et al., 1998; Konda & Tsitsiklis, 1999) has been widely adopted, since it combines the strengths of both value-based and policy-based RL approaches. When adopting the actor-critic framework, one needs to further find ways to handle the multi-objective structure. Toward this end, most of the recent MORL works (e.g., (Nguyen et al., 2020; Chen et al., 2021; Qiu et al., 2024; Zhou et al., 2024b; Ehrgott, 2005; Fliege et al., 2019)) can be categorized into two major classes: (1) Scalarization methods (e.g., linear scalarization (LS) and weighted-Chebyshev (WC)) that convert an MORL problem into a single-

Figure 1: The logic of our approach.

objective RL problem; and (2) adaptive gradient methods (e.g., MGDA method (Désidéri, 2012)) that aim to find a common improving direction.

Despite its significance, the theoretical foundation of the MORL problem still remains in its infancy. The major *limitations* of the existing MORL theories are three-folded: **(1)** Most empirical successes in MORL are built upon complex deep neural networks (DNN) (Yang et al., 2019; Nguyen et al., 2020; Chen et al., 2021). However, these proposed algorithms are typically of heuristic nature, and lack theoretical finite-time convergence rate or sample complexity guarantee. **(2)** Some recent studies have attempted to establish theoretical foundations of MORL (Qiu et al., 2024; Zhou et al., 2024b; Wang et al., 2024). However, their analysis heavily relies on the simple linear function approximations or tabular setups, which are inapplicable to the commonly used DNN-based MORL actor-critic framework mentioned earlier. **(3)** While some works guarantee the convergence to a Pareto stationary policy with linear approximations (Zhou et al., 2024b; Hairi et al., 2025), the problem of finding a Pareto optimal policy remains elusive, let alone systematically exploring the entire Pareto front that consists of all Pareto optimal policies.

The limitations outlined above underscore a substantial gap between the empirical success of DNN-based actor–critic MORL methods and the absence of a rigorous theoretical foundation for these algorithms. This naturally raises the following question:

**(Q)**: Can we develop efficient methods for MORL with DNN-based function approximation to 1) achieve Pareto optimality convergence globally and 2) explore the entire Pareto optimal front?

**2) Technical Challenges.** Answering the above question is highly non-trivial and necessitates addressing the following key challenges:

- While the current literature offers some insights into applying actor-critic methods for solving MORL problems, their theoretical analyses are mostly limited to linear critic approximations and the extension to the DNN-based actor-critic frameworks for MORL remains under-explored. With the complex computations introduced by the DNN component, whether it is possible to obtain finite-time convergence in MORL remains an open question.

- Even with the simpler linear function approximations, existing MORL only guarantee the convergence to Pareto stationary policies (which may be viewed as locally Pareto optimal), serving merely as a necessary condition for Pareto optimality. In contrast, identifying weakly Pareto optimal policies remains highly challenging, as many widely used techniques (e.g., MGDA-based MORL approaches (Zhou et al., 2024b; Hairi et al., 2025)) ensure convergence only to Pareto stationary policies, without providing any guarantees of global Pareto optimality.

- Even if a weakly Pareto optimal policy is obtained using DNN-based actor-critic method with finite-time convergence rate guarantee, it remains unclear whether the approach can incorporate different objective preferences to systematically explore the entire Pareto front.

**3) Main Contributions.** To overcome these challenges and to affirmatively answer the above question, we develop a DNN-based actor-critic MORL method, which not only guarantees global convergence to a Pareto optimal policy with finite-time convergence rate, but also systematically explores the entire weakly Pareto optimal front. Specifically, we summarize our contributions as follows:

(1) We show that, to achieve the global convergence to Pareto optimality in MORL policy design, the use of the weighted-Chebyshev (WC) scalarization technique is not only desirable, but also critical. Specifically, by converting a vector-valued MORL problem into a scalar-valued RL problem through the WC-scalarization, we are able to design Pareto optimal policies for the WC-scalarized RL problem with global convergence guarantee. In addition, we propose a smooth approximation of the WC-scalarized problem to address the non-smoothness challenge introduced by the "min-max" structure of the WC-scalarization.

Table 1: Comparison of Different Algorithms.

| Algorithm | Model | Convergence | Rate | Exploration |
|---|---|---|---|---|
| Qiu et al. (2024) | Tabular | Global | $\mathcal{O}(T^{-\frac{1}{2}})$ | ✓ |
| Zhou et al. (2024b) | Linear | Stationary | $\mathcal{O}(T^{-1})$ | ✗ |
| Wang et al. (2024) | Linear | Stationary | $\mathcal{O}(T^{-1})$ | ✗ |
| Hairi et al. (2025) | Linear | Stationary | $\mathcal{O}(T^{-1})$ | ✓ |
| Yang et al. (2019) | DNN | NA | - | ✗ |
| Chen et al. (2021)[‡] | DNN | NA | - | ✓ |
| **This Work** | **DNN** | **Global** | $\mathcal{O}(\mathbf{T^{-1}})$ | ✓ |

*Convergence*: whether the algorithm converges to Pareto stationary or optimal policies, or if no such guarantee exists. *Rate*: the convergence rate of the algorithm. *Exploration*: whether the algorithm can explore the entire Pareto front. ‡: They study a different setup and do consider multiple preferences.

(2) We develop a DNN-based actor-critic algorithm for the WC-scalarized RL problem, which enjoys a finite-time global convergence rate of $\mathcal{O}(1/T)$ to a Pareto optimal policy, where $T$ denotes the total number of iterations. Also, thanks to the one-to-one mapping between the solution sets of WC-scalarized RL problem and the Pareto front of the original MORL problem, our WC-based method achieves **global convergence** to **any** point on the Pareto front of the MORL problem. To our knowledge, these theoretical guarantees are the first of their kind in the literature.

(3) To validate our algorithm, we conduct extensive numerical experiments on both recommendation system training and multi-objective robotic simulation, which confirms that our algorithm can efficiently explore the weakly Pareto optimal front.

## 2 RELATED WORK

In this section, we summarize the related works in MORL and two closely related fields: single-objective actor-critic algorithms, and multi-objective optimization algorithms.

**1) Single-Objective Actor-Critic Algorithms:** The actor-critic framework, along with their variants have been one of the most widely used approaches in RL (Konda & Tsitsiklis, 1999; Peters & Schaal, 2008; Mnih et al., 2016). Besides their empirical successes, several works have also established rigorous theoretical finite-time convergence rate and sample complexity results (Xu et al., 2020; Qiu et al., 2021; Cayci et al., 2024; Tan et al., 2025). Moreover, recent works have begun to explore the DNN-based actor-critic algorithms (Wang & Hu, 2021; Gaur et al., 2024; Zhang et al., 2025; Ganesh et al., 2025). However, the theories of DNN-based actor-critic approaches for MORL remain largely missing in the literature.

**2) Multi-Objective Optimization Algorithms:** The history of Multi-Objective Optimization (MOO) problems dates back to (Sawaragi et al., 1985), and recent years have seen increasing development of MOO theories (Ehrgott, 2005; Gunantara, 2018; Sharma & Kumar, 2022). For example, the theoretical understanding for MOO approaches such as weighted-Chebyshev (WC, Momma et al. (2022); Lin et al. (2024)), multi-gradient descent algorithm (MGDA, Désidéri (2012); Xu et al. (2025)) have been established. However, when being applied in DNN-based actor-critic MORL, the theoretical convergence results of these MOO approaches remain unclear.

**3) MORL Algorithms:** Compared to the previous two areas, the theoretical studies on MORL only started in recent years. Although several MORL algorithms have been proposed (e.g., (Yang et al., 2019; Zhou et al., 2024b; Qiu et al., 2024; Wang et al., 2024; Hairi et al., 2025)), their theoretical convergence results remain poorly understood. In particular, most of the existing works failed to address at least one of the following aspects: 1) the use of DNN-based models, 2) providing finite-time global convergence guarantees, and 3) exploring the entire Pareto optimal front. In contrast, our algorithm significantly advances MORL by simultaneously addressing all of these technical barriers. To summarize, Table 1 highlights the strengths of our approach compared to existing methods. Due to space limitation, we relegate more related works in the literature of MORL to Appendix A.

## 3 MULTI-OBJECTIVE REINFORCEMENT LEARNING: A PRIMER

In this section, we will begin by formally formulating the MORL problem and providing several key definitions in MORL. Next, we will introduce the DNN-based actor-critic approach for MORL.

Finally, we propose a new smooth approximation of the WC-scalarized problem for solving DNN-based actor-critic MORL.

## 3.1 THE MORL PROBLEM

Consider an $m$-objective Markov decision process (MOMDP): $(\mathcal{S}, \mathcal{A}, \{r_i\}_{i=1}^m, \mathcal{P}, \gamma)$, where $\mathcal{S}$ is the state space, $\mathcal{A}$ is the action space, $\{r_i\}_{i=1}^m$ is the reward signal vector with $r_i : \mathcal{S} \times \mathcal{A} \to \mathbb{R}$ for the $i$-th objective, $\mathcal{P} : \mathcal{S} \times \mathcal{A} \times \mathcal{S} \to [0,1]$ is the transition kernel, and $\gamma \in (0,1)$ is the discount factor. Here, we adopt the so-called "restart transition kernel" that has been widely used in the literature (e.g., (Xu et al., 2020; Chen et al., 2022)). Specifically, $\mathcal{P}$ is defined as $\mathcal{P}(s, a, s') := \gamma \mathbb{P}(s'|s, a) + (1-\gamma)\mathbb{I}\{s' = s_0\}$, where $s_0$ denotes the initial state. An MORL policy is denoted by $\pi_\theta : \mathcal{S} \times \mathcal{A} \to [0,1]$, where $\theta$ represents its parameters. Hence, for each objective $i \in [m]$, we define the cumulative discounted reward for policy $\pi_\theta$ as $J_i(\theta) := \mathbb{E}_{\pi_\theta}\big[\sum_{t=0}^\infty \gamma^t r_{i,t}\big]$. The MORL problem can thus be formulated as:

$$\max_\theta \mathbf{J}(\theta) = \big[J_1(\theta), \dots, J_m(\theta)\big]^\top. \tag{1}$$

In MORL, the vector-valued objective is usually associated with a preference weight vector $p \in \Delta_m^+$, where $\Delta_m^+ := \{p \in \mathbb{R}^m : p \geq 0, \sum_{i=1}^m p_i = 1\}$ denotes the standard $m$-simplex, which represents potentially different attention on each objective. As mentioned earlier, since it is impossible to maximize multiple objective with a single policy $\pi_\theta$ in general, we introduce the following optimality criterion for solving MORL problems:

**Definition 1** (Pareto Optimality). Policy $\pi_\theta$ dominates $\pi_{\theta'}$ if and only if $J_i(\theta) \geq J_i(\theta'), \forall i \in [m]$, and $J_i(\theta) > J_i(\theta'), \exists i \in [m]$. Policy $\pi_\theta$ is Pareto optimal if no other policy $\pi_{\theta'}$ dominates $\theta$. Also, policy $\pi_\theta$ is weakly Pareto optimal if no other policy $\pi_{\theta'}$ satisfies: $J_i(\theta') > J_i(\theta), \forall i \in [m]$.

Clearly, Pareto optimality implies weak Pareto optimality, while the converse is not true. We can interpret Pareto optimality as the inability to find a policy that improves the performance of each objective simultaneously. Moreover, we also denote the set of all Pareto optimal (resp. weakly Pareto optimal) policies as $\Theta_\mathrm{P}$ (resp. $\Theta_\mathrm{WP}$), and the Pareto front (resp. weak Pareto front) as $\{F(\theta) : \theta \in \Theta_\mathrm{P}\}$ (resp. $\{F(\theta) : \theta \in \Theta_\mathrm{WP}\}$).

## 3.2 THE DEEP-NEURAL ACTOR-CRITIC APPROACH FOR MORL

Next, we will introduce the basics in the actor-critic approach, which is followed by the deep-neural actor-critic approach for MORL.

**1) The Actor-Critic Framework:** The actor-critic framework involves two stages: First, for some given policy, the critic component evaluates its value function, indicating the "goodness" of that policy; Second, based on the approximated value function, the actor component updates the policy using policy gradients. Specifically, the value function is defined as:

$$V_{\theta,i}(s) = \mathbb{E}\Big[\sum_{t=0}^\infty \gamma^t r_{i,t}\Big| s_0 = s, a_t \sim \pi_\theta(\cdot|s_t)\Big], \quad i \in [m].$$

However, the true value of $V_{\theta,i}(s)$ is unknown during the MORL process. Thus, the critic component approximates the state value function as $\widehat{V}(s; W_i)$ using techniques such as TD-learning, where $W_i$ denotes the parameters of the critic model. In this paper, $W_i$ is assumed to be parameterized by DNNs, which will be discussed later.

We can also define the advantage function as $\mathrm{Adv}_{\theta,i}(s, a) = r(s, a) + \gamma \mathbb{E}\big[V_{\theta,i}(s')|s' \sim \mathcal{P}(\cdot|s, a)\big] - V_{\theta,i}(s)$. The actor can compute the policy gradients by utilizing the following policy gradient theorem (Xu et al., 2020; Zhang et al., 2025):

**Lemma 1** (Policy Gradient Theorem). *Under the restart kernel $\mathcal{P}$, for any policy $\pi_\theta$ and for any $i \in [m]$, the gradient of $J_i(\theta)$ satisfies: $\nabla_\theta J_i(\theta) \propto \mathbb{E}\big[\nabla_\theta \log \pi_\theta(a|s)Adv_{\theta,i}(s, a)\big|(s, a) \sim \nu(\theta)\big]$, where $\nu(\theta)$ denotes the stationary distribution, which will be specified in Assumption 2.*

**2) Deep-Neural Actor-Critic Method for MORL:** As mentioned earlier, no theoretical foundations have been established for DNN-based actor-critic in the MORL literature. However, there do exist

some theoretical foundations for DNN-based actor-critic approaches in single-objective RL (Cai et al., 2019; Gaur et al., 2024; Zhang et al., 2025). Similar to these existing works, we also adopt multi-layer perceptron architecture for DNN-based actor-critic for MORL in this paper. Specifically, we encode each state $s$ by some $x \in \mathbb{R}^d$ with a one-to-one mapping, where it is assumed that $\|x\|_2 = 1$ without loss of generality. Then, the DNN can be represented as follows:

$$x^{(0)} = Ax, \qquad x^{(b)} = \frac{1}{\sqrt{w}}\text{Sigmoid}(W^{(b)}x^{(b-1)}), \forall b \in [D], \qquad y = c^\top x^{(D)},$$

where $A \in \mathbb{R}^{w \times d}$, $W^{(b)} \in \mathbb{R}^{w \times w}, \forall b \in [D]$, and $c \in \mathbb{R}^w$ are the parameters of the DNN, and $\text{Sigmoid}(\mathbf{v}) := \frac{1}{1+\exp^{-\mathbf{v}}}$ denotes the Sigmoid function. Notably, after initializing all entries of $A$ and $W^{(b)}, \forall b \in [D]$ independently following $\mathcal{N}(0, 2)$, and those in $c$ independently following $\mathcal{N}(0, 1)$, we only update $W = (W^{(1)}, \ldots, W^{(D)})$ during training. We thus simplify the notation $\widehat{V}(x; W, A, c)$ to $\widehat{V}(x; W)$. According to (Shen et al., 2022; Zhang et al., 2024), we have the following universal approximation result for the Sigmoid-DNN with depth $D$ and width $w$:

**Lemma 2** (Universal Approximation). *Suppose $V_{\theta,i}(x)$ is Lipschitz continuous (see Assumption 3). Then, there exists a Sigmoid-DNN parameterized by $W$, such that: $\max_{x,i,\theta} |\widehat{V}(x; W) - V_{\theta,i}(x)| = \widetilde{\mathcal{O}}(w^{-\frac{2}{d}} D^{-\frac{2}{d}})$, where $\widetilde{\mathcal{O}}(\cdot)$ hides the constants and logarithmic terms.*

Lemma 2 says that, when $V_{\theta,i}$ is Lipschitz continuous, as the width and depth of the DNN increase, the approximation error vanishes at nearly a rate of $1/(wD)^{\frac{2}{d}}$.

**Remark 1.** It is worth highlighting that the ReLU activation function is not compatible in our DNN-based MORL context. Due to the "scale-invariant" property of ReLU, i.e., $\alpha\text{ReLU}(v) = \text{ReLU}(\alpha v), \forall \alpha > 0$, the null space of the Fisher matrix introduced in Assumption 4 remains non-empty when ReLU is used in DNNs. Fortunately, this issue can be avoided by using alternative activation functions, such as Sigmoid, Tanh, and so on. According to Zhang et al. (2024), the universal approximation capabilities of DNNs with these activations are of the same order as ReLU-based DNNs in terms of width and depth.

### 3.3 A Smoothed Weighted-Chebyshev Method

Weighted-Chebyshev (WC) is a scalarization method for transforming a vector-valued optimization problem into a conventional scalar-valued optimization problem. Moreover, by varying the preference vector the $m$-dimensional standard simplex, one can systematically explore the entire weakly Pareto optimal front. To be conformal to the polarity of the standard WC-scalarization, we first transform the "reward maximization" in MORL in Eq. (1) into a "regret minimization" form. Let $J_i^{\text{ub}}$ denote an upper bound of $J_i(\theta)$. Then, we can reformulate the MORL problem as follows[1]:

$$\min_\theta \{\mathbf{J}^{\text{ub}} - \mathbf{J}(\theta)\} = [J_1^{\text{ub}} - J_1(\theta), \ldots, J_m^{\text{ub}} - J_m(\theta)]^\top,$$

where $J_i^{\text{ub}} - J_i(\theta) > 0, \forall i \in [m], \theta$. For any given preference vector $p \in \Delta_m^+$, the WC problem is defined as $\min_\theta g(\theta \mid p) := \|p \odot (\mathbf{J}^{\text{ub}} - \mathbf{J}(\theta))\|_\infty$, where $\odot$ denotes the Hadamard product. However, the WC-scalarization is in the "min-max" form, which is non-smooth and results in ill-defined gradient for the WC objective function. To address this challenge, we consider a smoothed WC-scalarization defined as follows (Lin et al., 2024):

$$\min_\theta G_\mu(\theta \mid p) := \mu \log \left( \sum_{i=1}^m \exp \frac{p_i(J_i^{\text{ub}} - J_i(\theta))}{\mu} \right), \tag{2}$$

where $\mu > 0$ is a tunable hyperparameter. It is shown in (Lin et al., 2024) that the smooth WC approximation can approximate the original WC-scalarization and maintain desirable properties:

**Lemma 3** (Pareto Front Reconstruction). *There exists some constant $\mu_0 > 0$, such that, for any fixed $\mu \in (0, \mu_0]$, the policy $\theta$ is a weakly Pareto optimal policy if and only if it is the solution of Eq. (2) under some preference $p \in \Delta_m^{++}$, where $\Delta_m^{++}$ is the positive standard $m$-simplex.*

---

[1]Note that this transformation does not lose any generality, as the Pareto fronts of these two problems have an one-to-one correspondence.

---

**Algorithm 1** DNN-based Actor-Critic for MORL

---

1: **Input:** step-size $\alpha_t$, initial parameters $\theta_0$, initial state $s_0$, preference $p$, $\mathbf{J}^{\text{ub}}$, and $\mu$.
2: **for** $t = 0, 1, \ldots, T - 1$ **do**
3:     Let $s_{t_0}, \{W_{i,t}\}_{i=1}^m$ be output of Algorithm 2.
4:     **for** $l = 0, 1, \ldots, M - 1$ **do**
5:         Observe: $s_{t_{l+1}}$ and $r_{i,t_{l+1}}, i \in [m]$.
6:         Sample: $a_{t_{l+1}} \sim \pi_{\theta_t}(\cdot | s_{t_{l+1}})$.
7:         Compute: $\psi_{t_l} = \nabla_\theta \log \pi_{\theta_t}(s_{t_l}, a_{t_l})$.
8:         **for** $i \in [m]$ **do**
9:             Compute: $\delta_{i,t_l} = \widehat{V}(s_{t_l}; W_{i,t}) - r_{i,t_{l+1}} - \gamma \widehat{V}(s_{t_{l+1}}; W_{i,t})$.
10:     **for** $i \in [m]$ **do**
11:         Compute: $\widehat{\nabla} J_i(\theta_t) = \frac{1}{M} \sum_{l=0}^{M-1} \delta_{i,t_l} \psi_{t_l}$.
12:         Compute: $\widehat{J}_i(\theta_t) = \widehat{V}(s_0; W_{i,t})$.
13:     Compute: $d_t$ according to Equation (3).
14:     Update: $\theta_{t+1} = \theta_t - \alpha_t \frac{d_t}{\|d_t\|}$.
15: **Output:** Policy $\theta_T$.

---

**Algorithm 2** DNN-Based Critic for MORL

---

1: **Input:** $s_0$, $\pi_{\theta_t}$, step-size $\beta$, iteration steps $K$, projection radius $B$.
2: **Initialize:** $\mathcal{B}(B) = \{W : \|W^{(b)} - W^{(b)}(0)\|_{\text{F}} \leq B, \forall h \in [D]\}$. $W_i(0) = W_i = W(0), \forall i \in [m]$.
3: **for** $k = 0, 1, \ldots, K - 1$ **do**
4:     Sample the tuple: $(s_k, a_k, \{r_{i,k+1}\}_{i=1}^m, s_{k+1}, a_{k+1})$, where $a_k \sim \pi_{\theta_t}(\cdot | s_k)$.
5:     **for** $i \in [m]$ **do**
6:         Compute TD-error: $\delta_{i,k} = \widehat{V}(s_k, ; W_i(k)) - r_{i,k+1} - \gamma \widehat{V}(s_{k+1}; W_i(k))$.
7:         Update: $\widetilde{W}_i(k+1) = W_i(k) - \beta \delta_{i,k} \cdot \nabla_W \widehat{V}(s_k; W_i(k))$.
8:         Project: $W_i(k+1) = \arg\min_{W \in \mathcal{B}(B)} \|W - \widetilde{W}_i(k+1)\|_2$.
9:         Update: $W_i = \frac{k+1}{k+2} W_i + \frac{1}{k+2} W_i(k+1)$.
10: **Output:** $s_{K-1}, \{W_i\}_{i=1}^m$.

---

This property highlights that solving the smoothed WC approximation problem to optimality and varying $p$ across the $m$-simplex still allow us to explore the entire weak Pareto front. We denote $\theta^*(p, \mu)$ as the minimizer of Eq. (2). Solving the MORL problem is then equivalent to solving this scalar-valued and smooth minimization problem, i.e., finding $\theta^*(p, \mu)$ for any $p \in \Delta_m^{++}$.

## 4 DNN-BASED ACTOR-CRITIC ALGORITHM FOR MORL

**1) Overview:** Our DNN-based actor-critic algorithm, presented in Algorithm 1, solves the smoothed WC-scalarized problem with a double-loop structure. In the inner-loop, the critic component iterates for $K$ steps, leveraging TD-learning method to approximate the $m$ value functions $V_{\theta,i}$ for the current policy $\theta$ with $m$ DNNs. The outer-loop executes $T$ rounds in total, where in each round, the actor component approximates the gradient of $G_\mu$ using the obtained value functions, and then updates the policy $\theta$, which is also parameterized by a DNN with the same width and depth.

**2) The Critic Component:** The critic component is presented in Algorithm 2, which aims to compute a value function approximation $\widehat{V}(x; W_i)$ for each objective $i \in [m]$. Specifically, for the current policy $\theta_t$ and the $i$-th objective, the critic first computes the TD-error $\delta_{i,k}$ at step $k$, then performs a TD update. The newly obtained parameters are then projected onto a ball centered at $W(0)$ with radius $B$, i.e., $\mathcal{B}(B)$. This projection ensures the non-expansive property of the convex ball, which is useful in the subsequent analysis.

**3) The Actor Component:** Each actor step $t$ begins with a Markov batch sampling with a batch-size of $M$. During this process, the algorithm maintains the score function $\psi_{t_l}$ and TD-error $\delta_{i,t_l}$ for each

$i \in [m]$. Upon the completion of Markov batch sampling, we approximate each objective function $J_i(\theta_t)$ and its gradient $\nabla J_i(\theta_t)$ according to Lemma 1, leading to the approximations $\widehat{J}_i(\theta_t)$ and $\widehat{\nabla} J_i(\theta_t)$, respectively. Then, we leverage these results to estimate the policy gradient $\nabla G_\mu(\theta_t \mid p)$ by substituting the ground truth with our approximations, as follows:

$$d_t = -\sum_{i=1}^{m} \frac{\exp\left(\frac{p_i(J_i^{\text{ub}} - \widehat{J}_i(\theta_t))}{\mu}\right)}{\sum_{i'=1}^{m} \exp\left(\frac{p_{i'}(J_{i'}^{\text{ub}} - \widehat{J}_{i'}(\theta_t))}{\mu}\right)} p_i \widehat{\nabla} J_i(\theta_t). \tag{3}$$

Finally, the policy $\theta_{t+1}$ is updated using a gradient descent step based on $G_\mu(\theta_t \mid p)$. Three important remarks are in order.

**Remark 2.** While our algorithm follows the actor-critic framework, the key novelty and difference stem from the policy update step. Specifically, after using policy gradient theorem (i.e., Lemma 1) to approximate $\widehat{\nabla} J_i(\theta_t)$, we do not directly perform a gradient descent step on $J_i(\theta_t)$. Instead, we utilize $J_i(\theta_t)$ to further approximate $\nabla G_\mu(\theta_t \mid p)$ and then perform a gradient update. This is because our new objective is the smoothed WC-scalarization of the MORL problem.

**Remark 3.** We note that in several existing MORL works, the actor component utilizes the MGDA technique (Zhou et al., 2024b; Wang et al., 2024; Hairi et al., 2025). In particular, after approximating $\nabla J_i(\theta_t)$ by $\widehat{\nabla} J_i(\theta_t)$, these works solve a quadratic programming $\min_\lambda \|\lambda \odot \widehat{\nabla} J_i(\theta_t)\|_2^2$ to determine a common descent direction. While these MGDA-based approaches enable finite-time convergence rate analysis (Désidéri, 2012), the inherent limitations of MGDA only guarantee a finite-time convergence rate result to a Pareto stationary policy, rather than a global convergence to a Pareto optimal policy. In contrast, through the smooth WC-scalarization problem (i.e., Problem (2), our proposed algorithm is able to exploit the special properties of the policy gradients combined with Lemma 3, which play a key role in ensuring global convergence to Pareto optimal policies and systematical Pareto exploration.

**Remark 4.** It is worth noting that our actor-critic framework employs $V$-function approximation (Chen et al., 2022; Hairi et al., 2022; Zhou et al., 2024b) rather than $Q$-function (Cai et al., 2019; Gaur et al., 2024; Zhang et al., 2025). This design offers significant benefits in practical implementation. To see this, note that in Line 12 of Algorithm 1, we need to approximate $\widehat{J}_i(\theta_t)$ for each objective $i$ (unique to the smooth WC-scalarized MORL approach and unseen in the previous literature). Using the $Q$-approximation $\widehat{J}_i(\theta_t) = \sum_a \widehat{Q}(s_0, a; W_{i,t}) \pi_{\theta_t}(a|s_0)$ requires enumerating the entire action space, which is impractical or even infeasible in MORL problems with large or continuous action space (e.g., video streaming recommendation systems and LLM alignment). In contrast, our $V$-approximation circumvents this difficulty and substantially enhances the capability and efficiency in solving the MORL problem.

## 5 THEORETICAL CONVERGENCE ANALYSIS

We begin by stating some useful assumptions in this section, which will be followed by our main theoretical results on finite-time global convergence and their further insights.

**Assumption 1** (Reward). There exists some $r_{\max} > 0$ such that, $r_{i,t} \in [0, r_{\max}], \forall t \geq 0, i \in [m]$.

**Assumption 2** (Geometric Mixing Time). For any policy $\pi_\theta$, there exists a stationary distribution $\nu(\theta)$ for $(s, a)$. Moreover, for any policy $\pi_\theta$, there exist positive constants $\eta$ and $\rho \in (0, 1)$ such that $\sup_{s \in \mathcal{S}} \|\mathbb{P}(s_t, a_t | s_0 = s) - \nu(\theta)\|_{\text{TV}} \leq \eta \rho^t, \forall t \geq 0$.

Assumption 2 is standard in the literature (Zou et al., 2019b; Xu et al., 2020; Gaur et al., 2024; Wang et al., 2024). Notably, the stationary distribution and the mixing behavior of the MOMDP can also be equivalently ensured by assuming the irreducible and aperiodic MOMDP (Hairi et al., 2022; Zhou et al., 2024b; Zhang et al., 2025).

**Assumption 3** (Lipschitz Continuity). $J_i(\theta)$ and $\nabla J_i(\theta)$ are Lipschitz continuous, i.e., there exist two positive constants $L_J$ and $M_J$ such that, for any $i \in [m]$, and for any $\theta$ and $\theta'$, we have:

$$|J_i(\theta) - J_i(\theta')| \leq L_J \|\theta - \theta'\|_2, \qquad \|\nabla J_i(\theta) - \nabla J_i(\theta')\|_2 \leq M_J \|\theta - \theta'\|_2.$$

Additionally, $V_{\theta,i}(x)$ is $L_V$-Lipschitz continuous, i.e., for any $x, x', i$ and $\theta$, we have:

$$|V_{\theta,i}(x) - V_{\theta,i}(x')| \leq L_V \|x - x'\|_2.$$

Assumption 3 is also commonly adopted in the literature (Wang et al., 2024; Zhou et al., 2024b; Gaur et al., 2024; Zhang et al., 2025), which i) allows us to apply the descent lemma in theoretical convergence analysis, and ii) ensures the universal approximation result presented in Lemma 2.

**Assumption 4.** For any policy $s$, $a$ and $\theta$, there exist positive a constant $M_g$ such that the score function satisfies: $\|\nabla_\theta \log \pi_\theta(a|s)\|_2 \leq M_g$.

**Assumption 5.** For any policy $\theta$ and for any $i \in [m]$, there exists positive constants $\sigma$ and $\epsilon_{\text{bias}}$ such that:

$$\mathbb{E}\big(\text{Adv}_{\theta,i}(s,a) - (1-\gamma)(F(\theta)^{-1}\nabla_\theta J_i(\theta))^\top \nabla_\theta \log \pi_\theta(a|s)\big)^2 \leq \epsilon_{\text{bias}},$$

where $(s,a) \sim \nu(\pi_i^*)$ (stationary distribution under the optimal policy $\pi_i^*$ with respect to objective $i$), $F(\theta) = \mathbb{E}_{\nu(\theta)}(\nabla_\theta \log \pi_\theta(a|s)\nabla_\theta \log \pi_\theta(a|s)^\top) + \sigma I$.

Assumptions 4 and 5 are also widely used in the existing works (Liu et al., 2020; Agarwal et al., 2021; Ding et al., 2022; Gaur et al., 2024; Zhang et al., 2025). Specifically, Assumption 5 is known as the "compatible function approximation" condition, which ensures that the policy function class (represented by DNNs in this paper) is sufficiently rich such that the advantage function $\text{Adv}(\cdot)$ can be well approximated by the score function $\psi(\cdot)$.

**Assumption 6.** For any $\theta \in \mathbb{R}^n$, and any $p \in \Delta_m^{++}$, the minimum singular value of $H(\theta \mid p)$ is strictly positive, i.e., there exists some $\delta_0 > 0$, such that, $\sigma_{\min}(H(\theta \mid p)) \geq \delta_0$, where:

$$H(\theta \mid p) = \Big[\nabla_\theta\big(p_1(J_1^{\text{ub}} - J_1(\theta))\big), \ldots, \nabla_\theta\big(p_m(J_m^{\text{ub}} - J_m(\theta))\big)\Big] \in \mathbb{R}^{n \times m}.$$

Assumption 6, which holds as long as $H(\theta \mid p)$ is column full rank, ensures that the gradient matrix is non-singular, making $\nabla G_\mu(\theta \mid p)$ tractable. With these assumptions, we are now ready to state our main results. Due to space limitation, we relegate the proof details to Appendix B.

**Theorem 1.** *Suppose all the assumptions hold. When selecting $\mu$ to be small enough, $\alpha_t = \frac{\alpha}{t}$, $\alpha \geq \max\{1, \frac{M_g L_J \sqrt{m}}{\sigma \delta_0} + \frac{\mu \log m \sqrt{m}}{\delta_0}\}$, $\beta = 1/\sqrt{K}$, $w = \Omega(d^3 D^{-\frac{11}{2}})$, $B = \Theta(w^{\frac{1}{32}}D^{-6})$, and $K = \Omega(D^4)$, with probability at least $1 - \exp^{-\Omega(\log^2 w)}$, Algorithm 1 achieves the following global convergence guarantee for any $p \in \Delta_m^{++}$:*

$$\mathbb{E}\Big[G_\mu(\theta_T \mid p) - G_\mu(\theta^*(p,\mu) \mid p)\Big] = \mathcal{O}\left(\frac{1}{T}\right) + \mathcal{O}\left(\sqrt{\epsilon_{bias}}\right) + \mathcal{O}\left(M^{-\frac{1}{2}}\right)$$

$$+ \mathcal{O}\left(w^{-\frac{2}{d}}D^{-\frac{2}{d}}\mu^{-1}m^{\frac{1}{2}}\right) + \widetilde{\mathcal{O}}\left(w^{\frac{1}{32}}D^{-\frac{7}{2}}K^{-\frac{1}{4}}\mu^{-1}m^{\frac{1}{2}}\right) + \widetilde{\mathcal{O}}\left(w^{-\frac{1}{24}}D^{-4}\mu^{-1}m^{\frac{1}{2}}\right).$$

**Corollary 1.** *For any $\epsilon > 0$, in order to achieve an $\epsilon$-optimal solution, i.e., to ensure $\mathbb{E}[G_\mu(\theta_T \mid p) - G_\mu(\theta^*(p,\mu) \mid p)] \leq \epsilon$, we can select $T = \Omega(\epsilon^{-1})$, $M = \Omega(\epsilon^{-2})$, and $K = \Omega(m^{\frac{1}{2}}\epsilon^{-1})$. Then, the corresponding sample complexity is $T(M + K) = \mathcal{O}(m^{\frac{1}{2}}\epsilon^{-3})$.*

**Remark 5.** Theorem 1 says that Algorithm 1 efficiently solves Equation (2), and converges to its global minimum at a rate of $\mathcal{O}(1/T)$. Moreover, by Lemma 3, we know that achieving this global minimum for Eq. (2) implies obtaining a weakly Pareto optimal policy for the original MORL problem. Furthermore, as mentioned earlier, Algorithm 1 also explores the entire weak Pareto front $\Theta_{\text{WP}}$ by varying the preference vector $p$ in the positive standard $m$-simplex $\Delta_m^{++}$. To our knowledge, our results on finite-time global convergence, sample complexity, and Pareto front reconstruction are the first of their kind in the DNN-based actor-critic literature for MORL.

**Remark 6.** Additionally, Theorem 1 not only establishes the theoretical foundation for the DNN-based actor-critic for MORL for the first time, but also provides interesting insights into how DNNs affect performance. Specifically, increasing the depth $D$ of the DNNs significantly improves the performance of Algorithm 1, whereas changes in width $w$ have a negligible impact.

**Remark 7.** As shown in Appendix B, a key step in achieving global convergence in our analysis is to verify the *performance difference lemma* proposed by (Kakade & Langford, 2002) compatible with our problem context. This property, primarily applied in the single-objective scenario, heavily relies on the well-defined gradient of the objective function. By utilizing the smoothed WC-scalarization, we are indeed able to derive an smooth, scalar-valued objective function, and provide a variant of this performance difference lemma tailored for MORL.

Table 2: Comparison of our method with baseline methods.

| Algorithm | Click↑ | Like↑(e-2) | Follow↑(e-4) | Comment↑(e-3) | Forward↑(e-3) | Dislike↓(e-4) | WatchTime↑ |
|---|---|---|---|---|---|---|---|
| Behavior-Clone | 0.534 | 1.231 | 4.608 | 3.225 | **1.119** | 2.304 | 1.285 |
| MOAC (Linear approx.) | 0.535 0.30% | 1.261 2.46% | 4.946 7.33% | 2.780 −13.8% | 1.105 −1.23% | 1.395 −39.4% | 1.249 −2.84% |
| MOCHA (Linear approx.) | 0.535 0.15% | 1.348 9.48% | 4.109 −10.8% | 3.033 −5.97% | 1.020 −8.86% | 1.373 −40.4% | 1.235 −3.94% |
| PDPG (DNN) | **0.539** **1.02**% | 1.228 −0.26% | 4.828 4.78% | 3.165 −1.86% | 0.919 −17.8% | **1.140** **−50.5**% | **1.308** **1.74**% |
| **Ours** (DNN) | **0.539** **1.02**% | **1.372** **11.47**% | **5.042** **9.42**% | **3.324** **3.07**% | 0.960 −14.19% | 1.538 −33.24% | 1.293 0.58% |

# 6 NUMERICAL EXPERIMENTS

## 6.1 SIMULATION ON RECOMMENDATION SYSTEMS

**1) Experimental Setup:** We conduct experiments on Kuairand dataset (Gao et al., 2022), an unbiased sequential dataset collected from the recommendation logs of a video-sharing app. It provides multiple potentially conflicting signals, which makes it especially useful for evaluating MORL methods. To compare with existing algorithms, we evaluate algorithm performances on optimizing 7 main feedback signals: Click, Like, Follow, Comment, Forward, Dislike, and Watch Time. We compare our approach with four MORL baselines: PDPG (Chen et al., 2021), MOAC (Zhou et al., 2024b), and MOCHA (Hairi et al., 2022). Due to space limitation, the details are relegated to Appendix C.1.

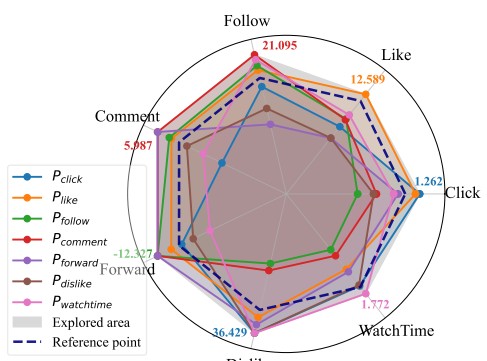

Figure 2: Pareto front boundary discovery.

**2) Experimental Results:** We summarize our experiment results in Table 2, where Behavior-Clone, a supervised method that mimics real customer behavior, serves as a benchmark for all MORL methods. We observe that (i) in general, the utilization of DNN significantly enhances overall performance, highlighting the strength of DNN over linear approximation; (ii) as DNN-based methods, our method outperforms PDPG on metric *Like*, *Follow*, and *Comment*, while PDPG exhibits better results on *Dislike* and *Watch Time*. To evaluate our method on Pareto front boundary discovery, we set a group of 7 preference vectors, each exhibiting a maximal preference toward a specific objective. The result is shown in Fig. 2, where the reference point is exactly the output of our algorithm using the preference vector used in Table 2. Due to space limitation, the additional numerical results, including the measurements of Hypervolume and the $\epsilon$-metric are relegated to Appendix C.1.

## 6.2 EXPERIMENTS ON BI-OBJECTIVE ROBOTIC SIMULATION

**1) Experimental Setup:** We also validate our algorithm on a robotic simulation task within MuJoCo-Walker-2d-v5 environment (Felten et al., 2023). Here, we consider a bi-objective problem, where the *Move* objective encourages the walker to move forward, while the *Control* objective aims to minimize the control effort. To validate the systematical Pareto exploration capability of our approach, we randomly sample various preference vectors across the standard simplex $\Delta_2^+$. In addition, we also conduct ablation studies within this Walker environment to demonstrate how our algorithm outperforms the linear scalarization-based actor-critic method. Due to space limitation, the detailed setups are relegated to Appendix C.2.

Figure 3: Visualization of Pareto fronts achieved by our algorithm and linear scalarization-based actor-critic method.

**2) Experimental Results:** In Fig. 3, each point represents the rewards of two objectives under a fixed preference vector $p$. The blue curve, which can be interpreted as the Pareto front obtained by our method, clearly illustrates the trade-off between these two objectives. This demonstrates the efficiency of our algorithm in Pareto exploration. In contrast, the yellow curve shows the front achieved by the linear scalarization-based actor-critic method. Clearly, it is dominated by the blue curve, highlighting the effectiveness of WC technique used in our algorithm. Due to space limitations, we relegate the additional results on how the size of DNNs impacts the algorithm to Appendix C.2.

## 7 CONCLUSION

We studied the MORL problem in this paper and proposed a DNN-based actor-critic algorithm utilizing the smoothed weighted-Chebyshev (WC) technique. Our algorithm achieves global optimality and facilitates systematical Pareto front exploration. Moreover, we proved that the algorithm converges to the global optima at rate of $\mathcal{O}(1/T)$ along with a sample complexity of $\mathcal{O}(m^{\frac{3}{2}}\epsilon^{-3})$, establishing the first theoretical guarantees for DNN-based actor-critic approaches in MORL. Numerous experiments on recommendation system training and multi-objective robotic simulation further verified the efficiency of our proposed algorithm.

## ETHICS STATEMENT

We confirm that The Code of Ethics has been carefully reviewed, and this paper fully adheres to the ICLR Code of Ethics. This work presents no potential societal consequences. Hence, we deem it unnecessary to highlight any specific aspects herein.

## REPRODUCIBILITY STATEMENT

We confirm that this work is reproducible. Specifically, the theories presented in this paper are clearly demonstrated with necessary assumptions and detailed proofs. Besides, the experimental setups and datasets utilized are thoroughly detailed in the appendix.

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

APPENDIX

## A  ADDITIONAL RELATED WORK ON MORL

In addition to the related works discussed in Section 2, we present additional literature in this section. We primarily focus on two more branches of MORL methods and also include works that provide practical guidance for MORL.

**Geometric and Pareto analysis-based Approaches.** In addition to scalarization-based MORL algorithms, there are works that focus on the geometric properties of the Pareto front and develop approaches inspired by the concept of Pareto optimality (Ye & Liu, 2022; Zhang et al., 2023; Shu et al., 2024; Liu et al., 2025; Zhao et al., 2025; Song et al., 2025). However, these works either do not provide theoretical guarantees for finite-time convergence or only allow convergence to stationary policies.

**Evolutionary-based Approaches.** Evolutionary approaches (Moriarty et al., 1999; Bai et al., 2023) are also widely used in the field of reinforcement learning. In recent years, several revolutionary MORL algorithms have been proposed (Khadka et al., 2019; Zhou et al., 2024a; Adde et al., 2025; Zhao et al., 2025; Janmohamed & Cully, 2025). These methods adopt a population-based approach to explore the Pareto front in parallel, which improves the efficiency of the algorithms to some extent. However, they remain heuristic and lack foundational theories to precisely characterize their performance.

**Measurements in MORL.** Several works also provide guidance on the metrics and benchmarks used in MORL (Van Moffaert et al., 2013; Roijers et al., 2018; Hayes et al., 2021; Felten et al., 2023). Metrics such as HyperVolume and the $\epsilon$-metric are also employed in this paper to numerically verify our algorithm.

## B  THEORETICAL PROOF OF THEOREM 1

*Proof.* The proof can be divided into three steps. First, we apply the descent lemma to control the dynamic $G_\mu(\theta_t \mid p)$ for each $t$, leading to an iteration result. Second, we demonstrate that the approximations in Algorithm 1 carefully control each term in the obtained result. Finally, we combine all the components to complete the analysis.

**Step A. Apply Descent Lemma and Iterate on $G_\mu(\theta \mid p)$.**

We begin by stating the following important lemma.

**Lemma 4.** *For any $\kappa \geq 0$, and for any $p \in \Delta_m^{++}$, $\mu$, denoting $C(p, \mu) = \frac{M_g}{\sigma} L_J + \mu \log m + \max_i \left( p_i (J_i^{ub} - J_i(\theta^*(p, \mu)) + \frac{\sqrt{\epsilon_{bias}}}{1-\gamma}) \right)$, we have:*

$$G_\mu(\theta \mid p) \leq C(p, \mu),$$

*where we simply denote $\| \cdot \|_2$ as $\| \cdot \|$ in the sequel.*

*Proof.* According to Kakade & Langford (2002); Ding et al. (2022); Gaur et al. (2024); Zhang et al. (2025), under Assumptions 4 and 5, for any policy $\theta$ and any $i \in [m]$, it holds that:

$$J_i(\theta^*(p, \mu)) - J_i(\theta) \leq \frac{\sqrt{\epsilon_{bias}}}{1-\gamma} + \frac{M_g}{\sigma} \|\nabla_\theta J_i(\theta)\|.$$

Thus, denoting $h_i(\theta \mid p) = p_i f_i(\theta)$, we can reformulate this result to further obtain:

$$J_i(\theta^*(p, \mu)) - J_i^{ub} + J_i^{ub} - J_i(\theta) \leq \frac{\sqrt{\epsilon_{bias}}}{1-\gamma} + \frac{M_g}{\sigma} \|\nabla_\theta J_i(\theta)\|,$$

$$\implies f_i(\theta) - \frac{M_g}{\sigma} \|\nabla f_i(\theta)\| \leq J_i^{ub} - J_i(\theta^*(p, \mu)) + \frac{\sqrt{\epsilon_{bias}}}{1-\gamma},$$

$$\implies h_i(\theta \mid p) - \frac{M_g}{\sigma} \|\nabla h_i(\theta \mid p)\| \leq p_i \left( J_i^{ub} - J_i(\theta^*(p, \mu)) + \frac{\sqrt{\epsilon_{bias}}}{1-\gamma} \right).$$

On the one hand, according to Assumption 3, we know that $\|\nabla h_i(\theta \mid p)\| \le p_i L_J, \forall i \in [m], \theta$. On the other hand, the property of "Log-Sum" inequality ensures that $G_\mu(\theta \mid p) \le \max_i (p_i f_i(\theta)) + \mu \log m$. Hence, we combine these results to get:

$$
\begin{aligned}
G_\mu(\theta \mid p) &\le \max_i \left(p_i f_i(\theta)\right) + \mu \log m \\
&= \max_i h_i(\theta \mid p) + \mu \log m \\
&\le \max_i \left( \frac{M_g}{\sigma} \|\nabla h_i(\theta \mid p)\| + p_i \left( J_i^{\mathrm{ub}} - J_i(\theta^*(p,\mu)) + \frac{\sqrt{\epsilon_{\mathrm{bias}}}}{1-\gamma} \right) \right) + \mu \log m \\
&\le \frac{M_g}{\sigma} L_J + \mu \log m + \max_i \left( p_i \left( J_i^{\mathrm{ub}} - J_i(\theta^*(p,\mu)) + \frac{\sqrt{\epsilon_{\mathrm{bias}}}}{1-\gamma} \right) \right) \\
&\le C(p,\mu),
\end{aligned}
$$

holds for any $\kappa \ge 0$, which ends the proof. $\qquad\square$

Notably, this is an extension of the *performance difference lemma* (Kakade & Langford, 2002), indicating that the objective function $G_\mu(\theta \mid p)$ derived from smooth WC technique still retains this important property: we can quantify the optimality of any policy $\theta$ by evaluating the value $C(p,\mu)$.

Now, we consider the sequence $\{\theta_t\}_{t=0}^{T-1}$, where $\theta_{t+1} = \theta_t - \alpha_t \frac{d_t}{\|d_t\|}$ by Algorithm 1. According to descent lemma, we have:

$$
\begin{aligned}
G_\mu(\theta_{t+1} \mid p) &\le G_\mu(\theta_t \mid p) + \nabla G_\mu(\theta_t \mid p)^\top (\theta_{t+1} - \theta_t) + \frac{L_\mu}{2} \|\theta_{t+1} - \theta_t\|^2 \\
&= G_\mu(\theta_t \mid p) - \alpha_t \frac{\nabla G_\mu(\theta_t \mid p)^\top d_t}{\|d_t\|} + \frac{L_\mu \alpha_t^2}{2} \\
&\le G_\mu(\theta_t \mid p) - \alpha_t \|\nabla G_\mu(\theta_t \mid p)\| + 2\alpha_t \|d_t - \nabla G_\mu(\theta_t \mid p)\| + \frac{L_\mu \alpha_t^2}{2},
\end{aligned}
$$

where the last inequality can be derived from the following argument with $v = \nabla G_\mu(\theta_t \mid p) - d_t$ and $d = d_t$:

$$
\begin{aligned}
& 0 \le \|v\| \cdot \|d\| + v^\top d, \\
\iff\ & \|v\| \cdot \|d\| + \|d\|^2 \le \|d\|^2 + 2\|v\| \cdot \|d\| + v^\top d, \\
\implies\ & \|d\| \cdot \|v + d\| \le \|d\|^2 + 2\|v\| \cdot \|d\| + v^\top d, \\
\iff\ & -\frac{(v+d)^\top d}{\|d\|} \le 2\|v\| - \|v + d\|.
\end{aligned}
$$

Then, according to Assumption 6, we have the following result:

$$
\nabla G_\mu(\theta \mid p) = \sum_{i=1}^m \frac{\exp\left(\frac{h_i(\theta|p)}{\mu}\right)}{\sum_{i'=1}^m \exp\left(\frac{h_{i'}(\theta|p)}{\mu}\right)} \nabla h_i(\theta \mid p) = \left( H(\theta \mid p) \right) \lambda(\theta, p, \mu),
$$

$$
\implies \|\nabla G_\mu(\theta \mid p)\| \ge \sigma_{\min}\left( H(\theta \mid p) \right) \|\lambda(\theta, p, \mu)\| \ge \frac{\delta_0}{\sqrt{m}},
$$

where we denote:

$$
\lambda_i(\theta, p, \mu) = \frac{\exp\left(\frac{h_i(\theta|p)}{\mu}\right)}{\sum_{i'=1}^m \exp\left(\frac{h_{i'}(\theta|p)}{\mu}\right)}, \quad \lambda(\theta, p, \mu) = (\lambda_1(\theta, p, \mu), \dots, \lambda_m(\theta, p, \mu))^\top.
$$

Hence, we apply Lemma 4 to further get:

$$
G_\mu(\theta \mid p) - \kappa \|\nabla G_\mu(\theta \mid p)\| \le C(p,\mu) - \kappa \|\nabla G_\mu(\theta \mid p)\| \le C(p,\mu) - \kappa \frac{\delta_0}{\sqrt{m}}.
$$

Thus, the aforementioned result obtained by the descent lemma can be handled as follows:

$$G_\mu(\theta_{t+1} \mid p)$$

$$\leq G_\mu(\theta_t \mid p) - \alpha_t \|\nabla G_\mu(\theta_t \mid p)\| + 2\alpha_t \|d_t - \nabla G_\mu(\theta_t \mid p)\| + \frac{L_\mu \alpha_t^2}{2}$$

$$\leq \left(1 - \frac{\alpha_t}{\kappa}\right) G_\mu(\theta_t \mid p) + 2\alpha_t \|d_t - \nabla G_\mu(\theta_t \mid p)\| + \frac{L_\mu \alpha_t^2}{2} + \frac{\alpha_t}{\kappa}\left(C(p,\mu) - \kappa \frac{\delta_0}{\sqrt{m}}\right),$$

which, according to Fatkhullin et al. (2023); Gaur et al. (2024); Zhang et al. (2025), and by selecting $\alpha_t = \frac{\alpha}{t}$ and $\alpha \geq \kappa$, further implies:

$$G_\mu(\theta_{t+1} \mid p) \leq \frac{1}{t} G_\mu(\theta_2 \mid p) + \frac{2\alpha}{t} \sum_{\tau=2}^{t} \|d_\tau - \nabla G_\mu(\theta_\tau \mid p)\| + \frac{C(p,\mu) - \kappa \frac{\delta_0}{\sqrt{m}}}{\kappa} + \frac{L_\mu \alpha^2}{2t}.$$

Note that the property of "Log-Sum" inequality also ensures that $\max_i \left(p_i f_i(\theta)\right) \leq G_\mu(\theta \mid p)$. Hence, when selecting $\kappa = \max\{1, \frac{M_g L_J \sqrt{m}}{\sigma \delta_0} + \frac{\mu \log m \sqrt{m}}{\delta_0}\}$, we have:

$$\frac{C(p,\mu) - \kappa \frac{\delta_0}{\sqrt{m}}}{\kappa} = \frac{\frac{M_g}{\sigma} L_J + \mu \log m + \max_i \left(p_i(J_i^{\text{ub}} - J_i(\theta^*(p,\mu)) + \frac{\sqrt{\epsilon_{\text{bias}}}}{1-\gamma})\right)}{\kappa} - \frac{\delta_0}{\sqrt{m}}$$

$$\leq \frac{1}{\kappa} \cdot \left(G_\mu(\theta^*(p,\mu) \mid p) + \frac{\sqrt{\epsilon_{\text{bias}}}}{1-\gamma} + \frac{M_g}{\sigma} L_J + \mu \log m\right) - \frac{\delta_0}{\sqrt{m}}$$

$$\leq G_\mu(\theta^*(p,\mu) \mid p) + \frac{\sqrt{\epsilon_{\text{bias}}}}{1-\gamma}.$$

Finally, we combine these results to get:

$$G_\mu(\theta_{t+1} \mid p) - G_\mu(\theta^*(p,\mu) \mid p)$$

$$\leq \frac{1}{t} G_\mu(\theta_2 \mid p) + \frac{2\alpha}{t} \sum_{\tau=2}^{t} \|d_\tau - \nabla G_\mu(\theta_\tau \mid p)\| + \frac{L_\mu \alpha^2}{2t} + \frac{\sqrt{\epsilon_{\text{bias}}}}{1-\gamma}. \tag{4}$$

Then, we need to control each term in the RHS of Equation (4) to guarantee the convergence performance.

**Step B. Control $\|d_t - \nabla G_\mu(\theta_t \mid p)\|$.**

For each $t$, we first add and subtract one term as follows:

$$\|d_t - \nabla G_\mu(\theta_t \mid p)\|^2 \leq \underbrace{2\|d_t - H(\theta_t \mid p)\widehat{\lambda}(\theta_t, p, \mu)\|^2}_{A_t} + \underbrace{2\|H(\theta_t \mid p)\widehat{\lambda}(\theta_t, p, \mu) - \nabla G_\mu(\theta_t \mid p)\|^2}_{B_t}.$$

We also introduce the following notations:

$$\widehat{\lambda}_i(\theta, p, \mu) = \frac{\exp\left(\frac{p_i(\widehat{J}_i^{\text{ub}} - \widehat{J}_i(\theta))}{\mu}\right)}{\sum_{i'=1}^{m} \exp\left(\frac{p_{i'}(\widehat{J}_{i'}^{\text{ub}} - \widehat{J}_{i'}(\theta))}{\mu}\right)}, \quad \widehat{\lambda}(\theta, p, \mu) = (\widehat{\lambda}_1(\theta, p, \mu), \ldots, \widehat{\lambda}_m(\theta, p, \mu))^\top.$$

Therefore, for $A_t$, we have:

$$A_t \overset{\flat}{=} 2\|\sum_{i=1}^{m} \widehat{\lambda}_i(\theta_t, p, \mu) \cdot p_i\left(\nabla J_i(\theta_t) - \widehat{\nabla} J_i(\theta_t)\right)\|^2$$

$$\overset{\dagger}{\leq} 2 \sum_{i=1}^{m} \widehat{\lambda}_i(\theta_t, p, \mu)\|p_i\left(\nabla J_i(\theta_t) - \widehat{\nabla} J_i(\theta_t)\right)\|^2$$

$$\overset{\ddagger}{\leq} 2 \max_i \|\widehat{\nabla} J_i(\theta_t) - \nabla J_i(\theta_t)\|^2,$$

where $\flat$ is due to the definition of $d_t$ according to Equation (3), $\dagger$ is due to $\widehat{\lambda}(\theta_t, p, \mu) \in \Delta_m^{++}$ and convexity of $\|\cdot\|^2$, and $\ddagger$ is due to $p \in \Delta_m^{++}$.

In order to further bound $A_t$, we leverage the following fact according to Cai et al. (2019); Zhang et al. (2025) to show the optimality of critic after update. Notably, the original results are primarily established for the $Q$ function, whereas it is not difficult to verify that the same arguments also yield parallel results for the $V$ function.

**Fact 1** (Cai et al. (2019))**.** *Let locally linear approximated V-function for any parameter $W$ be:*

$$\widehat{V}_0(x; W) = \widehat{V}(x; W(0)) + \langle \nabla_W \widehat{V}(x; W(0)), (W - W(0)) \rangle,$$

*and corresponding TD-error be:*

$$\delta_0(x, r, x'; W) = \widehat{V}_0(x; W) - r - \gamma \widehat{V}_0(x'; W).$$

*If $W^*$ satisfies:*

$$\mathbb{E}_{s \sim \nu} \left( (\delta_0(x, r, x'; W^*) \cdot \langle \nabla_W \widehat{V}_0(x; W^*)), (W - W^*) \rangle \right) \geq 0, \forall W \in \mathcal{B}(B),$$

*where, with a slight abuse of notation, $\nu$ here denotes the stationary distribution for state space (under the transition of the restart kernel and some policy $\pi_\theta$), then we say $W^*$ is a stationary point (since there is no descent direction at $W^*$). Then, for any $A$, $W(0)$ and $c$, there exists some stationary point $W^*$, and $\widehat{V}_0(\cdot; W^*)$ is the unique, global optimum of the minimization problem for policy $\theta$:*

$$\min_W \mathbb{E}_{x \sim \nu(\theta)} \left[ (\widehat{V}(x; W) - \Pi_{\mathcal{F}_{B,w}} \mathcal{T}^{\pi_\theta} \widehat{V}(x; W))^2 \right],$$

*where $\mathcal{F}_{B,w} = \{\widehat{V}(x; W(0)) + \langle \nabla_W \widehat{V}(x; W(0)), (W - W(0)) \rangle : W \in \mathcal{B}(B)\}$, and $\Pi_{\mathcal{F}_{B,w}}$ denotes the projection operation to the function class $\mathcal{F}_{B,w}$.*

Thus, for each $i \in [m]$, we can reformulate the desired term as:

$$\|\widehat{\nabla} J_i(\theta_t) - \nabla J_i(\theta_t)\|^2$$
$$\leq \underbrace{3\|\widehat{\nabla} J_i(\theta_t) - \widehat{\nabla} J_i(\theta_t; \delta(W_{i,t}^*))\|^2}_{A_{t,1}} + \underbrace{3\|\widehat{\nabla} J_i(\theta_t; \delta(W_{i,t}^*)) - \widehat{\mathrm{Adv}}(\theta_t; \delta(W_{i,t}^*))\|^2}_{A_{t,2}} + \underbrace{3\|\widehat{\mathrm{Adv}}(\theta_t; \delta(W_{i,t}^*)) - \nabla J_i(\theta_t)\|^2}_{A_{t,3}},$$

where:

$$\widehat{\nabla} J_i(\theta_t; \delta(W_{i,t}^*)) := \frac{1}{M} \sum_{l=0}^{M-1} \delta_{i,t_l}(W_{i,t}^*) \psi_{t_l},$$

$$\widehat{\mathrm{Adv}}(\theta_t; \delta(W_{i,t}^*)) := \mathbb{E} \Big( \mathrm{Adv}(s, a; W_{i,t}^*) \psi_{\theta_t}(s, a) \Big),$$

where $\delta_{i,t_l}(W_{i,t}^*) = \widehat{V}(s_{t_l}; W_{i,t}^*) - r_{i,t_{l+1}} - \gamma \widehat{V}(s_{t_{l+1}}; W_{i,t}^*)$ adopts the stationary $W_{i,t}^*$ for policy $\theta_t$, and $\psi_{\theta_t}(s, a) = \nabla_\theta \log(\pi_{\theta_t}(s, a))$. Hence, we next show that $A_{t,1}$, $A_{t,2}$, $A_{t,3}$ and $B_t$ can be controlled, respectively.

**Step B.1.** For $A_{t,1}$, we have:

$$A_{t,1} = 3\|\frac{1}{M} \sum_{l=0}^{M-1} \delta_{i,t_l} \psi_{t_l} - \frac{1}{M} \sum_{l=0}^{M-1} \delta_{i,t_l}(W_{i,t}^*) \psi_{t_l}\|^2$$

$$= 3\|\frac{1}{M} \sum_{l=0}^{M-1} \Big( \delta_{i,t_l} - \delta_{i,t_l}(W_{i,t}^*) \Big) \psi_{t_l}\|^2$$

$$\leq 3 \max_l \|\big( \delta_{i,t_l} - \delta_{i,t_l}(W_{i,t}^*) \big) \psi_{t_l}\|^2$$

$$\leq 3 M_g^2 \max_l \big( \delta_{i,t_l} - \delta_{i,t_l}(W_{i,t}^*) \big)^2,$$

where the last inequality is due to Assumption 4. For each $i \in [m]$, we can further get:

$$\left| \delta_{i,t_l} - \delta_{i,t_l}(W_{i,t}^*) \right|$$

$$= \left| \widehat{V}(s_{t_l}; W_{i,t}) - \gamma \widehat{V}(s_{t_{l+1}}; W_{i,t}) - \widehat{V}(s_{t_l}; W_{i,t}^*) + \gamma \widehat{V}(s_{t_{l+1}}; W_{i,t}^*) \right|$$

$$\leq \left| \widehat{V}(s_{t_l}; W_{i,t}) - \widehat{V}(s_{t_l}; W_{i,t}^*) \right| + \gamma \left| \widehat{V}(s_{t_{l+1}}; W_{i,t}) - \widehat{V}(s_{t_{l+1}}; W_{i,t}^*) \right|,$$

where the equality is due to the definition of TD-errors, and the inequality comes from the triangle inequality. Then, we take expectation, and follow the parallel results in Cai et al. (2019); Zhang et al. (2025) to get that:

$$\mathbb{E} \left[ |\delta_{i,t_l} - \delta_{i,t_l}(W_{i,t}^*)| \right] = \mathcal{O} \left( (BD^{\frac{5}{2}} K^{-\frac{1}{4}} + B^{\frac{4}{3}} w^{-\frac{1}{12}} D^4) \cdot \log^{\frac{3}{2}} w \log^{\frac{1}{2}} K \right),$$

holds with probability at least $1 - \exp(-\Omega(\log^2 w))$, when selecting $\beta = 1/\sqrt{K}$, $w = \Omega(d^3 D^{-\frac{11}{2}})$, $B = \Theta(w^{\frac{1}{32}} D^{-6})$, and $K = \Omega(D^4)$. Then, this implies:

$$\mathbb{E}(A_{t,1}) = \mathcal{O} \left( (B^2 D^5 K^{-\frac{1}{2}} + B^{\frac{8}{3}} w^{-\frac{1}{6}} D^8) \cdot \log^3 w \log K \right). \tag{5}$$

**Step B.2.** For $A_{t,2}$, we have:

$$\frac{A_{t,2}}{3} = \left\| \frac{1}{M} \sum_{l=0}^{M-1} \delta_{i,t_l}(W_{i,t}^*) \psi_{t_l} - \mathbb{E} \left( \mathrm{Adv}(s,a; W_{i,t}^*) \psi_{\theta_t}(s,a) \right) \right\|^2$$

$$= \frac{1}{M^2} \sum_{l=0}^{M-1} \left\| \delta_{i,t_l}(W_{i,t}^*) \psi_{t_l} - \mathbb{E} \left( \mathrm{Adv}(s,a; W_{i,t}^*) \psi_{\theta_t}(s,a) \right) \right\|^2$$

$$+ \frac{1}{M^2} \sum_{u \neq v} \langle \delta_{i,t_u}(W_{i,t}^*) \psi_{t_u} - \mathbb{E} \left( \mathrm{Adv}(s,a; W_{i,t}^*) \psi_{\theta_t}(s,a) \right), \delta_{i,t_v}(W_{i,t}^*) \psi_{t_v} - \mathbb{E} \left( \mathrm{Adv}(s,a; W_{i,t}^*) \psi_{\theta_t}(s,a) \right) \rangle.$$

We take expectation on both sides. Then, We first control the first term in the last equation as follows. For each $l \in \{0, \ldots, M-1\}$, we have:

$$\mathbb{E} \left\| \delta_{i,t_l}(W_{i,t}^*) \psi_{t_l} - \mathbb{E} \left( \mathrm{Adv}(s,a; W_{i,t}^*) \psi_{\theta_t}(s,a) \right) \right\|^2$$

$$\leq 2\mathbb{E} \left[ \| \delta_{i,t_l}(W_{i,t}^*) \psi_{t_l} \|^2 + \| \mathbb{E} \left( \mathrm{Adv}(s,a; W_{i,t}^*) \psi_{\theta_t}(s,a) \right) \|^2 \right]$$

$$\leq 2\mathbb{E} \left[ |\delta_{i,t_l}(W_{i,t}^*)|^2 \cdot \| \psi_{t_l} \|^2 \right] + 2\mathbb{E} \left[ |\mathrm{Adv}(s,a; W_{i,t}^*)|^2 \cdot \| \psi_{\theta_t}(s,a) \|^2 \right]$$

$$\leq 2M_g^2 \mathbb{E} |\delta_{i,t_l}(W_{i,t}^*)|^2 + 2M_g^2 \mathbb{E} |\mathrm{Adv}(s,a; W_{i,t}^*)|^2$$

$$\leq 4M_g^2 \max_l \mathbb{E} |\delta_{i,t_l}(W_{i,t}^*)|^2$$

$$= 4M_g^2 \left( (\frac{1+\gamma}{1-\gamma} + 1) r_{\max} + 2\widetilde{\mathcal{O}} \left( w^{-\frac{2}{d}} D^{-\frac{2}{d}} \right) \right)^2$$

$$= \mathcal{O}(1),$$

where the second last equation is due to the definition of TD-errors and Lemma 2, and the last equation holds because of the bounded reward. Then, we consider the second term. Without loss of generality, we assume $u < v$. Besides, we consider taking expectations conditioned on the filtration $\mathcal{F}_t$, where $\mathcal{F}_t$ denotes the samples up to iteration $t$. According to Hairi et al. (2022); Zhou et al. (2024b) the following results hold due to Assumption 2:

$$\mathbb{E} \left[ \langle \delta_{i,t_u}(W_{i,t}^*) \psi_{t_u} - \mathbb{E} \left( \mathrm{Adv}(s,a; W_{i,t}^*) \psi_{\theta_t}(s,a) \right), \delta_{i,t_v}(W_{i,t}^*) \psi_{t_v} - \mathbb{E} \left( \mathrm{Adv}(s,a; W_{i,t}^*) \psi_{\theta_t}(s,a) \right) \rangle \Big| \mathcal{F}_t \right]$$

$$= 2 \left( (\frac{1+\gamma}{1-\gamma} + 1) r_{\max} + 2\widetilde{\mathcal{O}} \left( w^{-\frac{2}{d}} D^{-\frac{2}{d}} \right) \right)$$

$$\cdot \mathbb{E} \left[ \left\| \mathbb{E} \left( \mathrm{Adv}(s_{t,l_v}, a_{t,l_v}; W_{i,t}^*) \psi_{\theta_t}(s,a) \big| \mathcal{F}_{t,l_u} \right) - \mathbb{E} \left( \mathrm{Adv}(s,a; W_{i,t}^*) \psi_{\theta_t}(s,a) \right) \right\| \Big| \mathcal{F}_t \right]$$

$$= 2M_g \left( (\frac{1+\gamma}{1-\gamma} + 1) r_{\max} + 2\widetilde{\mathcal{O}} \left( w^{-\frac{2}{d}} D^{-\frac{2}{d}} \right) \right)^2 \cdot \eta \rho^{v-u}.$$

Hence, we can further get:

$$\mathbb{E}(A_{t,2}) = \mathcal{O}\left(\frac{1}{M}\right) + \mathcal{O}\left(\frac{1}{M^2} \cdot M \frac{\eta\rho}{1-\rho}\right) = \mathcal{O}\left(\frac{1}{M}\right). \tag{6}$$

**Step B.3.** For $A_{t,3}$, according to Lemma 2, we have:

$$
\begin{aligned}
A_{t,3} &= \left\| \mathbb{E}\Big(\mathrm{Adv}(s,a;W_{i,t}^*)\psi_{\theta_t}(s,a)\Big) - \mathbb{E}\Big(\mathrm{Adv}_{\theta_t,i}(s,a)\psi_{\theta_t}(s,a)\Big) \right\|^2 \\
&= \left\| \mathbb{E}\Big[\big(\mathrm{Adv}(s,a;W_{i,t}^*) - \mathrm{Adv}_{\theta_t,i}(s,a)\big)\psi_{\theta_t}(s,a)\Big] \right\|^2 \\
&\le M_g^2 \Big(\mathbb{E}\big|\mathrm{Adv}(s,a;W_{i,t}^*) - \mathrm{Adv}_{\theta_t,i}(s,a)\big|\Big)^2 \\
&= M_g^2 \Big(\mathbb{E}\big|\gamma\mathbb{E}(\widehat{V}(s;W_{i,t}^*)) - \widehat{V}(s;W_{i,t}^*) - \gamma\mathbb{E}(V_{\theta_t,i}(s)) + V_{\theta_t,i}(s)\big|\Big)^2 \\
&\le 4M_g^2 \Big(\mathbb{E}\big|\widehat{V}(s;W_{i,t}^*) - V_{\theta_t,i}(s)\big|\Big)^2 \\
&\le 4M_g^2 \mathbb{E}\big(\widehat{V}(s;W_{i,t}^*) - V_{\theta_t,i}(s)\big)^2 \\
&= \widetilde{\mathcal{O}}\left(w^{-\frac{4}{d}}D^{-\frac{4}{d}}\right).
\end{aligned}
\tag{7}
$$

**Step B.4.** As for $B_t$, we can obtain:

$$
\begin{aligned}
B_t &= 2\|H(\theta_t \mid p)\big(\widehat{\lambda}(\theta_t,p,\mu) - \lambda(\theta_t,p,\mu)\big)\|^2 \\
&= 2\Big\| \sum_{i=1}^m \big(\widehat{\lambda}_i(\theta_t,p,\mu) - \lambda_i(\theta_t,p,\mu)\big) \cdot p_i \nabla J_i(\theta_t)\Big\|^2 \\
&\overset{\dagger}{\le} 2\sum_{i=1}^m p_i \big\|\big(\widehat{\lambda}_i(\theta_t,p,\mu) - \lambda_i(\theta_t,p,\mu)\big) \cdot \nabla J_i(\theta_t)\big\|^2 \\
&\overset{\ddagger}{\le} 2L_J^2 \sum_{i=1}^m p_i \big(\widehat{\lambda}_i(\theta_t,p,\mu) - \lambda_i(\theta_t,p,\mu)\big)^2 \\
&\le 2L_J^2 \max_i \big(\widehat{\lambda}_i(\theta_t,p,\mu) - \lambda_i(\theta_t,p,\mu)\big)^2.
\end{aligned}
$$

where $\dagger$ is due to $p \in \Delta_m^{++}$ and convexity of $\|\cdot\|^2$, and $\ddagger$ is due to Assumption 3.

This implies that, for any $i \in [m]$, we need to consider $\big(\widehat{\lambda}_i(\theta_t,p,\mu) - \lambda_i(\theta_t,p,\mu)\big)^2$. To this end, we first consider the following bias arising from approximations:

$$
\begin{aligned}
&\mathbb{E}\big|\widehat{J}_i(\theta_t) - J_i(\theta_t)\big| \\
=&\mathbb{E}\left|\widehat{V}(s_0;W_{i,t}) - V_{\theta_t,i}(s_0)\right| \\
\le&\mathbb{E}\left|\widehat{V}(s_0;W_{i,t}) - \widehat{V}(s_0;W_{i,t}^*)\right| + \mathbb{E}\left|\widehat{V}(s_0;W_{i,t}^*) - V_{\theta_t,i}(s_0)\right| \\
=&\mathcal{O}\left((BD^{\frac{5}{2}}K^{-\frac{1}{4}} + B^{\frac{4}{3}}w^{-\frac{1}{12}}D^4) \cdot \log^{\frac{3}{2}} w \log^{\frac{1}{2}} K\right) + \widetilde{\mathcal{O}}\left(w^{-\frac{2}{d}}D^{-\frac{2}{d}}\right).
\end{aligned}
$$

Let $\mathcal{J}_i(\theta) = p_i \frac{J_i^{\mathrm{ub}} - J_i(\theta)}{\mu}$ and $\widehat{\mathcal{J}}_i(\theta) = p_i \frac{J_i^{\mathrm{ub}} - \widehat{J}_i(\theta)}{\mu}$. Then, we can get:

$$\mathbb{E}\big|\widehat{\mathcal{J}}_i(\theta_t) - \mathcal{J}_i(\theta_t)\big| = \mathcal{O}\left((BD^{\frac{5}{2}}K^{-\frac{1}{4}} + B^{\frac{4}{3}}w^{-\frac{1}{12}}D^4) \cdot \log^{\frac{3}{2}} w \log^{\frac{1}{2}} K\mu^{-1}\right) + \widetilde{\mathcal{O}}\left(w^{-\frac{2}{d}}D^{-\frac{2}{d}}\mu^{-1}\right).$$

Then, we denote $\mathcal{J}(\theta) = (\mathcal{J}_1(\theta), \ldots, \mathcal{J}_m(\theta))^\top$, and $\widehat{\mathcal{J}}(\theta) = (\widehat{\mathcal{J}}_1(\theta), \ldots, \widehat{\mathcal{J}}_m(\theta))^\top$. For convenience, we also denote $\phi_i(\mathcal{J}(\theta)) = \frac{\exp(\mathcal{J}_i(\theta))}{\sum_j \exp(\mathcal{J}_j(\theta))}$. Then, according to Mean value theorem, we

know that there exists some $c \in \mathbb{R}^m$, such that:

$$
\left| \phi_i(\mathcal{J}(\theta)) - \phi_i(\widehat{\mathcal{J}}(\theta)) \right|^2 = \left| \nabla \phi_i(c)^\top \left( \mathcal{J}(\theta) - \widehat{\mathcal{J}}(\theta) \right) \right|^2
$$
$$
\leq \| \nabla \phi_i(c) \|_1^2 \| \mathcal{J}(\theta) - \widehat{\mathcal{J}}(\theta) \|_\infty^2
$$
$$
\leq \frac{1}{4} \max_i \left| \widehat{\mathcal{J}}_i(\theta_t) - \mathcal{J}_i(\theta_t) \right|^2,
$$

which implies that:

$$
\mathbb{E}\left( \widehat{\lambda}_i(\theta_t, p, \mu) - \lambda_i(\theta_t, p, \mu) \right)^2
$$
$$
= \mathbb{E}\left( \phi_i(\mathcal{J}(\theta_t)) - \phi_i(\widehat{\mathcal{J}}(\theta_t)) \right)^2
$$
$$
\leq \mathbb{E}\left[ \frac{1}{4} \max_i \left( \widehat{\mathcal{J}}_i(\theta_t) - \mathcal{J}_i(\theta_t) \right)^2 \right]
$$
$$
\leq \frac{1}{4} \mathbb{E}\left[ \sum_{i=1}^m \left( \widehat{\mathcal{J}}_i(\theta_t) - \mathcal{J}_i(\theta_t) \right)^2 \right]
$$
$$
= \mathcal{O}\left( (B^2 D^5 K^{-\frac{1}{2}} + B^{\frac{8}{3}} w^{-\frac{1}{6}} D^8) \cdot \log^3 w \log K \mu^{-2} m \right) + \widetilde{\mathcal{O}}\left( w^{-\frac{4}{d}} D^{-\frac{4}{d}} \mu^{-2} m \right),
$$

holds with probability at least $1 - \exp(-\Omega(\log^2 w))$ when selecting parameters as shown before. These results indicate that:

$$
\mathbb{E}(B_t) = \mathcal{O}\left( (B^2 D^5 K^{-\frac{1}{2}} + B^{\frac{8}{3}} w^{-\frac{1}{6}} D^8) \cdot \log^3 w \log K \mu^{-2} m \right) + \widetilde{\mathcal{O}}\left( w^{-\frac{4}{d}} D^{-\frac{4}{d}} \mu^{-2} m \right). \quad (8)
$$

**Step C. Complete the Proof.**

Combining Equations (5) to (8), we know that the following result holds with probability at least $1 - \exp(-\Omega(\log^2 w))$ when selecting parameters as shown before:

$$
\| d_t - \nabla G_\mu(\theta_t \mid p) \| = \mathcal{O}\left( (BD^{\frac{5}{2}} K^{-\frac{1}{4}} + B^{\frac{4}{3}} w^{-\frac{1}{12}} D^4) \cdot \log^{\frac{3}{2}} w \log^{\frac{1}{2}} K \mu^{-1} m^{\frac{1}{2}} \right)
$$
$$
+ \widetilde{\mathcal{O}}\left( w^{-\frac{2}{d}} D^{-\frac{2}{d}} \mu^{-1} m^{\frac{1}{2}} \right) + \mathcal{O}\left( M^{-\frac{1}{2}} \right).
$$

Then, we substitute this back to Equation (4) to obtain the following result:

$$
G_\mu(\theta_{t+1} \mid p) - G_\mu(\theta^*(p, \mu) \mid p)
$$
$$
\leq \frac{1}{t} G_\mu(\theta_2 \mid p) + \frac{2\alpha}{t} \sum_{\tau=2}^t \| d_\tau - \nabla G_\mu(\theta_\tau \mid p) \| + \frac{L_\mu \alpha^2}{2t} + \frac{\sqrt{\epsilon_{\text{bias}}}}{1 - \gamma}
$$
$$
= \mathcal{O}\left( \frac{1}{t} \right) + \mathcal{O}\left( \sqrt{\epsilon_{\text{bias}}} \right) + \mathcal{O}\left( M^{-\frac{1}{2}} \right) + \widetilde{\mathcal{O}}\left( w^{-\frac{2}{d}} D^{-\frac{2}{d}} \mu^{-1} m^{\frac{1}{2}} \right)
$$
$$
+ \mathcal{O}\left( (BD^{\frac{5}{2}} K^{-\frac{1}{4}} + B^{\frac{4}{3}} w^{-\frac{1}{12}} D^4) \cdot \log^{\frac{3}{2}} w \log^{\frac{1}{2}} K \mu^{-1} m^{\frac{1}{2}} \right)
$$

Therefore, be selecting $\alpha_t = \frac{\alpha}{t}$, $\alpha \geq \max\{1, \frac{M_g L_J \sqrt{m}}{\sigma \delta_0} + \frac{\mu \log m \sqrt{m}}{\delta_0}\}$, $\beta = 1/\sqrt{K}$, $w = \Omega(d^3 D^{-\frac{11}{2}})$, $B = \Theta(w^{\frac{1}{32}} D^{-6})$, and $K = \Omega(D^4)$, with probability at least $1 - \exp^{-\Omega(\log^2 w)}$, Algorithm 1 has the following global convergence guarantee for any $p \in \Delta_m^{++}$:

$$
\mathbb{E}\left[ G_\mu(\theta_T \mid p) - G_\mu(\theta^*(p, \mu) \mid p) \right]
$$
$$
= \mathcal{O}\left( \frac{1}{T} \right) + \mathcal{O}\left( \sqrt{\epsilon_{\text{bias}}} \right) + \mathcal{O}\left( M^{-\frac{1}{2}} \right) + \widetilde{\mathcal{O}}\left( w^{-\frac{2}{d}} D^{-\frac{2}{d}} \mu^{-1} m^{\frac{1}{2}} \right)
$$
$$
+ \widetilde{\mathcal{O}}\left( w^{\frac{1}{32}} D^{-\frac{7}{2}} K^{-\frac{1}{4}} \mu^{-1} m^{\frac{1}{2}} \right) + \widetilde{\mathcal{O}}\left( w^{-\frac{1}{24}} D^{-4} \mu^{-1} m^{\frac{1}{2}} \right),
$$

which ends the proof.

$\square$

# C  SETUPS AND ADDITIONAL RESULTS OF NUMERICAL EXPERIMENTS

In this section, we provide the detailed setups of our numerical experiments along with some additional results.

## C.1  SIMULATION ON RECOMMENDATION SYSTEMS

**1) Detailed Setup.** We conduct experiments on the Kuairand dataset (Gao et al., 2022), an unbiased sequential recommendation dataset collected from the recommendation logs of a video-sharing mobile app. It provides rich feedback of 12 distinct signals (e.g. click, like, view time, follows, comments) together with timestamps, user and item features, and over 30 side-features. Its unbiased exposure mechanism makes it especially useful for evaluating debiasing and causal recommendation methods.

To compare with existing algorithms, we evaluate algorithm performances on optimizing 7 main feedback signals: click, like, follow, comment, forward, dislike, view time, and compare our approach with several baselines (Chen et al., 2021; Zhou et al., 2024b; Hairi et al., 2022). To ensure a fair comparison, for DNN based methods (i.e., Chen et al. (2021), our work), we implement them using the same network architecture of a 3-layer perceptron each with ReLU as activation function for both critic and actor networks. For methods utilizing linear approximation, i.e., Zhou et al. (2024b); Hairi et al. (2025), we keep both critic and actor networks as single linear layer for linear approximation. In addition, for methods with preference vector as input (i.e., Hairi et al. (2025), our work), we set a unified preference vector for all objectives. We note here that the reward optimization among objectives could still be biased even with unified preference vector, given that different feedback signals have very different density in this data, e.g., the density of signal "forward" is $0.076\%$ in all 7 signals so there is very limited customer feedback can be learned. Finally, we benchmark all methods on metric normalized capped importance sampling (NCIS) (Zou et al., 2019a).

The preference vector used for the preference-based algorithms (Ours, MOCHA) shown in Table 2 is $p = \frac{\lambda}{||\lambda||}$ where $\lambda = [10.0, 1.0, 1.0, 1.0, 0.01, 0.1, 10.0]^\top$. We select this "seemingly random" preference vector to achieve a relatively balanced performances across all objectives, since, as mentioned above, the distribution of the dataset is highly biased. This is also the preference vector used to obtain the reference point in Figure 2. Moreover, the (unnormalized) preference vectors used in Figure 2 are listed in Table 3.

Table 3: Preference vectors used in Figure 2.

| | |
|---|---|
| $P_{\text{click}}$ | $[100.0, 1.0, 1.0, 1.0, 0.01, 0.1, 10.0]^\top$ |
| $P_{\text{like}}$ | $[10.0, 100.0, 1.0, 1.0, 0.01, 0.1, 10.0]^\top$ |
| $P_{\text{follow}}$ | $[10.0, 1.0, 100.0, 1.0, 0.01, 0.1, 10.0]^\top$ |
| $P_{\text{comment}}$ | $[10.0, 1.0, 1.0, 100.0, 0.01, 0.1, 10.0]^\top$ |
| $P_{\text{forward}}$ | $[10.0, 1.0, 1.0, 1.0, 100.0, 0.1, 10.0]^\top$ |
| $P_{\text{dislike}}$ | $[10.0, 1.0, 1.0, 1.0, 0.01, 100.0, 10.0]^\top$ |
| $P_{\text{watchtime}}$ | $[10.0, 1.0, 1.0, 1.0, 0.01, 0.1, 100.0]^\top$ |

**2) Additional Results.** We also provide the numerical results on the metrics of Hypervolume and $\epsilon$-metric Hayes et al. (2021) as follows:

Table 4: Hypervolume results in recommendation system (comparison with baselines).

| Algorithm | Behavior-Clone | MOAC | MOCHA | PDPG | **Ours** |
|---|---|---|---|---|---|
| HyperVolume ($\uparrow$) | 0.054 | 0.094 | 0.083 | 0.138 | **0.223** |

The performance on Hypervolume for our algorithm and the baselines is shown in Table 4, in which the preference-based algorithms are still based on the preference $p = \frac{\lambda}{||\lambda||}$ where $\lambda = [10.0, 1.0, 1.0, 1.0, 0.01, 0.1, 10.0]^\top$. Clearly, our algorithm outperforms all other baselines, which directly confirms the effectiveness of our approach.

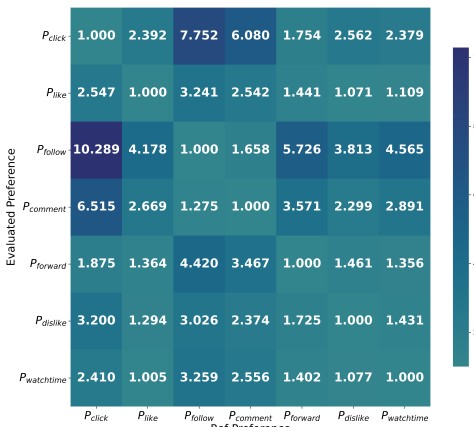

Figure 4: $\epsilon$-metric performances for different preference vectors.

We also provide the numerical results on the $\epsilon$-metric for the preference vectors listed in Table 3. As shown in Figure 4, except for the diagonal elements, all values are greater than 1, indicating that the trade-offs among conflicting objectives are well balanced by selecting distinct preference vectors. This further validates the systematic Pareto exploration capability of our algorithm.

### C.2 EXPERIMENTS ON BI-OBJECTIVE ROBOTIC SIMULATION

**1) Detailed Setup.** In MoJoCo-Walker-2d-v5 environment (Felten et al., 2023), we aim to control a walking robot (walker) to move forward. The basic setups are detailed as follows:

- **Episode.** In each episode, there are at most $T = 500$ time steps. Every episode ends once the walker falls down (referred to as the walker becoming "unhealthy") or when the maximum number of steps is reached, and a new episode is then initialized. Every experiment consists of a total of $1,000,000$ time steps.

- **State Space.** The state space has a dimension of 17, with each component representing either the position or velocity of a walker's body part. Among these, the first dimension, the position of the "z-coordinate", determines the walker's health: it's healthy only if this value lies within the interval $[0.8, 1.0]$.

- **Action Space.** The action space has a dimension of 6, with each component representing torque added to a specific body part of the walker. The transition kernel follows the laws of mechanics.

- **Reward Signals.** We define the following quantities: 1) "Health" outputs 1 if the robot remains standing, and 0 otherwise; 2) "Forward" denotes the velocity along the forward axis; and 3) "Cost" is proportional to $\|a\|_2^2$, i.e., the squared norm of the taken action $a$. We then consider two types of reward signals as follows. First, ***Move=Health+Forward*** represents the velocity along the forward axis, encouraging the walker to move forward quickly without falling. Second, ***Control=Health-Cost*** implies that the controller is also supposed to minimize the interference. Obviously, these two kinds of reward signals conflict with each other, motivating us to formulate this robotic simulation task as a bi-objective problem, in which we aim to maximize both objective functions.

- **Preference vectors.** In the experiments, we first randomly sample various preference vectors across the standard simplex $\Delta_2^+$ (Xu et al., 2020; Felten et al., 2024). After obtaining the Pareto front based on these preferences, we observe that the distribution of the resulting points is highly uneven across the front. Therefore, for the sparser regions, where the outputs for the related preferences are more sensitive, we manually select additional preferences to more precisely characterize the shape of the Pareto front. For instance, if the points obtained by $p_1 = [0.9, 0.1]^\top$ and $p_1 = [0.8, 0.2]^\top$ are far apart, we randomly sample additional preference vectors between them, such as $p' = [0.85, 0.15]^\top$, to fill the gap between them. For the ablation studies shown in Figure 5, we fix the preference vector to $p = [0.9, 0.1]^\top$, and modify other hyperparameters.

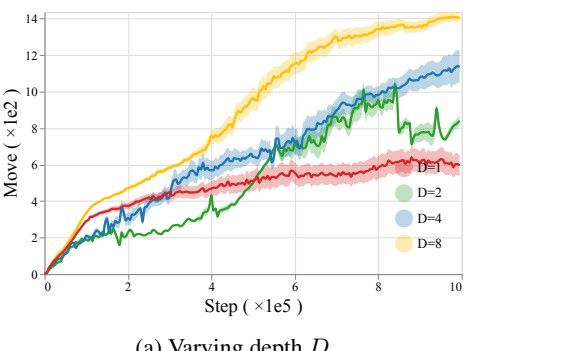
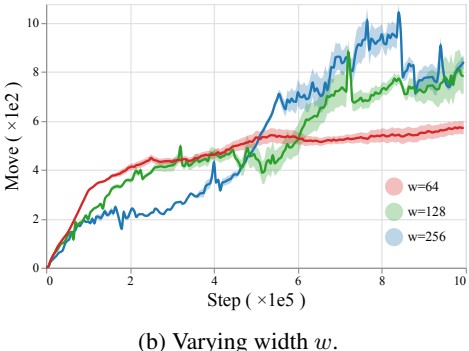

(a) Varying depth $D$.
        
(b) Varying width $w$.

Figure 5: Performance under different depth $D$ and hidden width $H$ under preference [0.9,0.1]. Each curve is smoothed using a moving average over 500 steps.

- **Others.** We also enumerate the default setting of other aforementioned parameters here. The discounted factor is set to $\gamma = 0.99$. The DNNs we utilized are with width of 256 and depth of 2. The smoothing parameter is set to $\mu = 0.05$. The upper bound is set to $\mathbf{J}^{\text{ub}} = \{2000, 1000\}$. The learning rates are set to $\alpha_t = \alpha = 3 \times 10^{-4}$, and $\beta = 10^{-3}$.

In addition to the Pareto exploration results, we also investigate how the parameters impact the performance of our approach. The detailed investigation setups are listed below:

- Depth $D$. We consider the DNN with different depths: $D \in \{1, 2, 4, 8\}$.
- Width $w$. We apply the DNN with different widths: $w \in \{64, 128, 256\}$.

**2) Additional Results.** We repeat each experiments with 3 different random seeds, and plot the mean and 95% confidence interval. Figure 5 provides the ablation studies by showing the forward reward curves under the default setup while varying a single hyperparameter.

As illustrated in Figure 5a, the reward returns consistently increase as the depth $D$ of the DNN increases. This suggests that the deeper DNN achieves better performance, which aligns with our theoretical results. Figure 5b considers the effect of the width $w$ of the DNN. While larger widths yield slightly better performance, the differences across settings are relatively not significant, which is also consistent with our theoretical analysis.

