# OpenReview forum: "Global Convergence and Pareto Front Exploration in Deep-Neural Actor-Critic Multi-Objective Reinforcement Learning"
_ICLR.cc/2026/Conference — Submitted to ICLR 2026_

### Official Review · Reviewer_E4KY · 2025-10-26

**Soundness:** 2
**Presentation:** 1
**Contribution:** 2
**Rating:** 4
**Confidence:** 3

**Summary:**

This paper introduces weighted Chebyshev techniques to reformulate the MORL problem, enabling the finding of weakly Pareto-optimal policies for the WC-scalarized RL formulation with a global convergence guarantee. It further proposes a smooth approximation of the WC-scalarized problem to address the non-smoothness issue, which makes neural network training feasible. Based on this, the authors also prove that a DNN-based actor–critic algorithm for the WC-scalarized RL problem enjoys finite-time global convergence. They provide experiments on a recommendation-system task and a multi-objective robotic simulation. Although the theoretical part is relatively complete and novel, the experimental evaluation and related algorithmic design have clear limitations.

**Strengths:**

1. The mathematical formulation and proofs are well structured and technically sound, making the theoretical contribution both clear and credible.
2. The proposed DNN-based implementation makes the method practical and easy to plug and play with state-of-the-art MORL approaches.
3. The paper provides experiments on both robotics and recommendation-system tasks, demonstrating the general applicability of the framework.

**Weaknesses:**

1. The experimental setup for the recommendation-system task is confusing:
(a) How many policies (and corresponding preference vectors) were trained in total? In addition, were all baselines evaluated using the same number of policies for a fair comparison?
(b) Could you clarify the rationale for selecting these baselines? Why were no state-of-the-art MORL methods included for comparison?
(c) What exactly does the statement "the reference point is our method on the same setting as in Table 2" mean? Please clarify how this reference point is defined and used in Figure 2.
(d) Why did not you use standard MORL evaluation metrics such as hypervolume or expected utility?
2. Regarding the robotics experiment, the appendix mentions that the authors just manually select a group of preferences. This choice is confusing and may have significantly limited the performance of both the proposed method and the baselines.
3. As noted above, the preference vectors in this paper are selected randomly. For experiments with a continuous Pareto front, this is impractical, as it cannot ensure adequate coverage of the front. However, the proposed method seems easily compatible with multi-policy MORL frameworks [1-3], which could systematically explore or sample preferences to better approximate the entire Pareto front.
4. The paper lacks an ablation study comparing the proposed WC scalarization with the standard linear scalarization, which would help verify whether the method indeed provides any practical advantage in exploring the Pareto front.

References
[1] Xu, Jie, et al. "Prediction-guided multi-objective reinforcement learning for continuous robot control." International conference on machine learning. PMLR, 2020.
[2] Felten, Florian, El-Ghazali Talbi, and Grégoire Danoy. "Multi-objective reinforcement learning based on decomposition: A taxonomy and framework." Journal of Artificial Intelligence Research 79 (2024): 679-723.
[3] Liu, Ruohong, et al. "Efficient Discovery of Pareto Front for Multi-Objective Reinforcement Learning." The Thirteenth International Conference on Learning Representations.

**Questions:**

Please refer to the questions listed in the Weakness section above. I would be happy to raise my score if the authors can address the concerns mentioned above.

---

> ### Author Response · Authors · 2025-11-23
> **Author Response - Part 1**
>
> > **Comment 1:** The experimental setup for the recommendation-system task is confusing: (a) How many policies (and corresponding preference vectors) were trained in total? In addition, were all baselines evaluated using the same number of policies for a fair comparison? (b) Could you clarify the rationale for selecting these baselines? Why were no state-of-the-art MORL methods included for comparison? \(c) What exactly does the statement "the reference point is our method on the same setting as in Table 2" mean? Please clarify how this reference point is defined and used in Figure 2. (d) Why did not you use standard MORL evaluation metrics such as hypervolume or expected utility?
>
> **Response:** Thanks for your comments and questions. We would like to respond your questions point-by-point as follows:
>
> **(a)** We only trained **one** policy for each algorithm, and our comparison is indeed fair. Specifically, given that some baselines are not preference-based (e.g., MOAC and PDPG), we evaluated preference-based methods (MOCHA and Ours) with a fixed preference vector $p = \frac{\lambda}{||\lambda||}$ where $\lambda = [10.0, 1.0, 1.0, 1.0, 0.01, 0.1, 10.0]^\top$, for objectives "Click," "Like," "Follow," "Comment," "Forward," "Dislike," and "WatchTime," respectively. The reason of such preference vector selection primarily lies in addressing the imbalanced data distribution in the Kuairand dataset. Both our method and MOCHA show balanced performance among objectives under this preference $p$ in the sense that no objective is severely penalized for limited improvements in other objectives, allowing a fair comparison between preference-based and preference-free algorithms.
>
> **(b)** Our rationale for selecting baselines is to include the **most relevant** works along two key dimensions: (i) they adopt the same MORL problem formulation, and (ii) their proposed algorithms provide theoretical performance guarantees. Also, as shown in **Section 2** and **Appendix A**, we have conducted a comprehensive literature review on MORL. However, most of these recent MORL works are either (i) not directly comparable because they adopt different problem settings, or (ii) heuristic methods that provide no theoretical performance guarantees. Thus, the baselines used in our experiments constitute the most appropriate and theoretically grounded benchmarks for a fair comparison.
>
>
> **\(c)** What we meant in the quoted sentence is that the reference point shown in Figure 2 is exactly the output of our algorithm using the preference vector $p$ mentioned in part (a), i.e., $p = \frac{\lambda}{||\lambda||}$ where $\lambda = [10.0, 1.0, 1.0, 1.0, 0.01, 0.1, 10.0]^\top$. Besides this, we have also selected different preference vectors in Figure 2 to validate the systematic Pareto exploration capability of our algorithm. To avoid similar confusions, we have rewritten this sentence and added the detailed setting in Lines 1160-1175 in our revision.
>
>
> **(d)** In our original submission, we used radar plots because they make it easy to directly visualize how our DNN-based actor–critic method balances the trade-offs among multiple conflicting objectives. That said, we do agree with the reviewer that using more systematic MORL evaluation metrics (e.g., Hypervolume and $\epsilon$-metric) can further enhance our experiments and performance evaluation. In this rebuttal period, we have conducted additional experiments using the Hypervolume metric and these new results are included in Appendix C.1 in our revision.

---

> ### Author Response · Authors · 2025-11-23
> **Author Response - Part 2**
>
> > **Comment 2:** Regarding the robotics experiment, the appendix mentions that the authors just manually select a group of preferences. This choice is confusing and may have significantly limited the performance of both the proposed method and the baselines.
>
> **Response:** Thanks for your comment. We would like to further clarify our preference selection as follows:
>
> **1) The original preference vectors are selected "nearly uniformly" across the standard simplex.** Although we select the preference vectors manually, they are chosen nearly uniformly within the standard simplex, i.e., we perform an almost uniform search of the simplex with a grid size of 0.1. The rationale behind our uniform grid selection is to cover a broad range of preference vectors, thereby validating our theoretical guarantee on systematic Pareto-front exploration.
>
> **2) An iterative preference vector selection process for further Pareto front exploration coverage.** We note that, despite the near-uniform selection of preference vectors, the resulting points on the Pareto front are not evenly distributed in general. Thus, the coverage of some regions of the Pareto front could be relatively sparse under the uniform grid selection for preference vectors. To address this limitation of uniform grid selection, during this rebuttal period, we have enhanced our preference vector selection as follows:
>
> We first uniformly at random sample various preference vectors across the standard two-dimensional simplex $\Delta_2^+$ (i.e., $p_1,p_2\in\Delta_2^+$ if $p_1+p_2=1$ and $p_1,p_2\ge 0$). After "sketching" the Pareto front based on these uniformly distributed preferences, if we observe that the distribution of the resulting points of vector-valued objectives are distributed unevenly across the front, then we will further select additional preference vectors for those sparsely covered regions, where the outputs for the corresponding preference vectors are more sensitive. For instance, if the points obtained by $p_1 = [0.9, 0.1]^\top$ and $p_1 = [0.8, 0.2]^\top$ are far apart, we again sample additional preference vectors between them uniformly at random, such as $p' = [0.85, 0.15]^\top$, to fill the gap between them.
>
> Again, we thank the reviewer's insightful comments regarding preference vector selections, which help us improve the quality and presentation of our experiments. We have incorporated the above changes in the revision (cf. Lines 1234-1241).

---

> ### Author Response · Authors · 2025-11-23
> **Author Response - Part 3**
>
> > **Comment 3:** As noted above, the preference vectors in this paper are selected randomly. For experiments with a continuous Pareto front, this is impractical, as it cannot ensure adequate coverage of the front. However, the proposed method seems easily compatible with multi-policy MORL frameworks [1-3], which could systematically explore or sample preferences to better approximate the entire Pareto front.
>
> > [1] Xu, Jie, et al. "Prediction-guided multi-objective reinforcement learning for continuous robot control." International conference on machine learning. PMLR, 2020.
>  [2] Felten, Florian, El-Ghazali Talbi, and Grégoire Danoy. "Multi-objective reinforcement learning based on decomposition: A taxonomy and framework." Journal of Artificial Intelligence Research 79 (2024): 679-723.
>  [3] Liu, Ruohong, et al. "Efficient Discovery of Pareto Front for Multi-Objective Reinforcement Learning." The Thirteenth International Conference on Learning Representations.
>
> **Response:** Thanks for the comments and suggestions. We would like to clarify our preference vector selections as follows:
>
> **1)** As mentioned in the previous response, our original preference vector selection process performs a nearly uniform grid search across the standard simplex, which is easy to implement. We do agree with the reviewer that the selected preferences may not be dense enough, and the coverage of Pareto front exploration may not be adequate. To address this, we have proposed a new iterative preference vector selection process for more comprehensive Pareto front coverage (see Lines 1233-1240 in our revision).
>
> **2)** After carefully reading the papers suggested by the reviewer, we note that:
>
> * All these papers select a large number of preference vectors to achieve sufficient Pareto-front coverage. However, given the limited time available during the rebuttal period, it is challenging for us to incorporate all of these preference-selection schemes into our framework. Instead, we propose a new iterative preference-vector selection scheme to enrich Pareto-front exploration. We hope the reviewer finds this approach satisfactory, and we would be happy to include additional results using those mentioned approaches with more time.
> * Even with a larger number of sampled points, their selection methods remain relatively simplistic, as they are not as "adaptive" as our new iterative preference-vector selection scheme. For instance, in [3], the authors either perform a manually uniform search (similar to our previous approach) or randomly sample preference vectors (which is more aligned with our current selection philosophy). We note that these approaches are, to a large extent, have been subsumed by our new iterative preference-vector selection scheme.
> * In all three papers, regardless of the number and density of the preference vectors, the resulting numerical presentation of the Pareto front remains discrete. On the other hand, as shown in these works, with a sufficient number of preference vectors, the Pareto front exploration coverage can be improved. As shown in our updated Figure 3, we have been able to improve our Pareto front exploration coverage in similar ways.
> * We thank the reviewer again for pointing out these references. We have incorporated all the above discussions and cited these works suggested by the reviewer in our revised manuscript (cf. Lines 1233-1240).
>
> ---
>
> > **Comment 4:** The paper lacks an ablation study comparing the proposed WC scalarization with the standard linear scalarization, which would help verify whether the method indeed provides any practical advantage in exploring the Pareto front.
>
> **Response:** Thanks for your comment. Per your suggestion, we have added the comparison results between our method and the linear scalarization (LS)-based method in **Section 6.2** in our revision. Based on these new results, we confirm that our WC-based algorithm consistently outperforms the LS-based method in terms of Pareto front exploration (cf. Figure 3).

---

> > ### Comment · Reviewer_E4KY · 2025-11-27
> >
> > Thank you for the clarifications. I appreciate the authors’ efforts during the rebuttal period. However, several concerns remain as follows:
> > 1. I am actually asking what specific preference vectors you used in Figure 2. You mention that 7 different preference vectors were used, each “maximal preference toward a specific objective,” but the numerical form of these vectors is not stated. If I understand correctly, for Figure 2, the preference vectors are one-hot–like vectors? In addition, for Table 2, I do not think using a single fixed preference vector is sufficient for assessing MORL algorithms.
> > 2. As a minor suggestion, I noticed that the authors added some additional experiments in the appendix. However, in the main text, this is only described vaguely as “we relegate additional results to Appendix.” For clarity and to better guide the reader, it would be helpful to explicitly mention what additional results are included—for example, stating that hypervolume metrics are provided in the appendix.
> > 3. Overall, the contribution of this paper is not enough. As a theoretical paper, the novelty appears incremental: Lemma 2, Lemma 3, and the smooth WC scalarization are largely based on prior work. Global convergence of DNN-based actor–critic methods has also been established in previous literature. The main contribution of this paper is therefore essentially combining smooth WC scalarization with an existing DNN-based actor–critic convergence framework for the MORL setting. As an experimental paper, the current evaluation is also not sufficiently comprehensive.
> >
> > Despite the incremental nature, I agree that establishing DNN-based convergence guarantees for MORL is conceptually important, given that modern RL methods rely heavily on deep actor–critic frameworks. I encourage the authors to strengthen the experimental section.

---

> ### Author Response · Authors · 2025-12-03
> **Follow-up Response - Part 1**
>
> > **Comment:** I am actually asking what specific preference vectors you used in Figure 2. You mention that 7 different preference vectors were used, each “maximal preference toward a specific objective,” but the numerical form of these vectors is not stated. If I understand correctly, for Figure 2, the preference vectors are one-hot–like vectors? In addition, for Table 2, I do not think using a single fixed preference vector is sufficient for assessing MORL algorithms.
>
> **Response:** Thanks for your follow-up question. We would like to further clarify the choice of preference vectors as follows:
>
> **1) Numerical Forms of the Preference Vectors:** In both Part \(c) in our previous response to your previous **Comment 1** and in Appendix C.1, we have provided the detailed settings of our numerical experiments on recommendation system. To further clarify, the preference vectors used in Figure 2 (before normalization) is listed in Table 3 (see Lines 1167-1175 in our revision) and also repeated below. These preference vectors are **not** one-hot vectors, even though each vector is heavily skewed toward a single objective. Technically, we select these *"heavily skewed"* preference vectors to *"explore the boundaries"* of the Pareto front.
>
>
> | Preference Vector | Value |
> | :---: | :--- |
> | $P_{\text{click}}$ | $[100.0, 1.0, 1.0, 1.0, 0.01, 0.1, 10.0]^\top$ |
> | $P_{\text{like}}$ | $[10.0, 100.0, 1.0, 1.0, 0.01, 0.1, 10.0]^\top$ |
> | $P_{\text{follow}}$ | $[10.0, 1.0, 100.0, 1.0, 0.01, 0.1, 10.0]^\top$ |
> | $P_{\text{comment}}$ | $[10.0, 1.0, 1.0, 100.0, 0.01, 0.1, 10.0]^\top$ |
> | $P_{\text{forward}}$ | $[10.0, 1.0, 1.0, 1.0, 100.0, 0.1, 10.0]^\top$ |
> | $P_{\text{dislike}}$ | $[10.0, 1.0, 1.0, 1.0, 0.01, 100.0, 10.0]^\top$ |
> | $P_{\text{watchtime}}$ | $[10.0, 1.0, 1.0, 1.0, 0.01, 0.1, 100.0]^\top$ |
>
> **2) The Reason for Using a Single Fixed Preference Vector:** We would like to clarify the reason why we use a single fixed preference vector in Table 2 as follows:
>
> * We note that some MORL baselines are "preference-free methods", meaning **no**  preference information is used as input. As a result, preference-free methods typically converge (if convergence happens at all) to an arbitrary Pareto solution. In contrast, preference-based methods leverage preference information to guide the convergence toward different points across the Pareto front, which holds the potential to enable systematic Pareto front exploration (but Pareto front exploration remains highly challenging even with the use of preference information). Our DNN-based actor-critic algorithm for MORL is a preference-based method, and one of our main contributions is that it offers the capability to systematically explore the entire Pareto front.
>
>   Therefore, to ensure a fair comparison between preference-free and preference-based methods, we use a **single fixed preference vector** in our preference-based method to obtain only one Pareto-optimal solution, rather than allowing multiple preference vectors to explore a set of Pareto-optimal solutions. As shown in Table 2, under the control of a specific preference vector, our **single-preference-based** approach outperforms the preference-free baselines across the majority of the objectives.

---

> ### Author Response · Authors · 2025-12-03
> **Follow-up Response - Part 2**
>
> > **Comment:** As a minor suggestion, I noticed that the authors added some additional experiments in the appendix. However, in the main text, this is only described vaguely as “we relegate additional results to Appendix.” For clarity and to better guide the reader, it would be helpful to explicitly mention what additional results are included—for example, stating that hypervolume metrics are provided in the appendix.
>
> **Response:** Thanks for your suggestion. We agree with the reviewer that a more detailed description would be helpful to guide the readers to understand the additional experiments. Following your suggestions, we have updated Section 6 as follows:
> * *"Due to space limitation, the additional numerical results, including the measurements of Hypervolume and the $\epsilon$-metric are relegated to Appendix C.1."* (cf. Lines 467-468)
> * *"Due to space limitations, we relegate the additional results on how the size of DNNs impacts the algorithm to Appendix C.2."* (cf. Lines 491-492)

---

> ### Author Response · Authors · 2025-12-03
> **Follow-up Response - Part 3**
>
> > **Comment:** Overall, the contribution of this paper is not enough. As a theoretical paper, the novelty appears incremental: Lemma 2, Lemma 3, and the smooth WC scalarization are largely based on prior work. Global convergence of DNN-based actor–critic methods has also been established in previous literature. The main contribution of this paper is therefore essentially combining smooth WC scalarization with an existing DNN-based actor–critic convergence framework for the MORL setting. As an experimental paper, the current evaluation is also not sufficiently comprehensive.
>
> **Response:** Thanks for your comments. To further clarify our contributions, in what follows, we will again outline the major technical challenges in this paper and then explain why our contributions go far beyond *"combining smooth WC scalarization with existing DNN-based actor-critic framework"* and instead introduces significant new ideas:
>
> 1. **WC-Scalarization Not Applicable in DNN-based Actor-Critic Framework:** Although the conventional weighted Chebyshev (WC)-scalarization technique is known for systematic Pareto frotn exploration in multi-objective optimization (MOO), it is **fundamentally unfit** for actor-critic policy gradient algorithm design due to its **non-smooth min-max** nature, leading to not well-defined policy gradients.
> 2. **Lack of a Theoretical Foundation for DNN-based Actor-Critic RL Framework:** Theoretical convergence and sample complexity analysis for deep-neural-network (DNN)-based actor-critic reinforcement learning with **nonlinear function approximation** are far more challenging and complex compared to actor-critic MORL methods based on **linear** function approximations. To date, there remains a lack of theoretical foundation on performance analysis for DNN-based actor-critic RL framework, which necessitates new proof techniques.
> 3. **No Known Results Exist for Establishing Global Pareto-Optimality Convergence:** Most exisiting theoretical works on reinforcement learning in the literature are limited to stationary convergence guarantee, rather than global convergence guarantee. Moreover, for MOO and MORL problems, global optimality in the Pareto sense (i.e., Pareto-optimality) is **fundamentally different** from the conventional notion of optimality in scalar-valued problems. To date, how to establish global Pareto-optimality convergence for DNN-based actor-critic MORL methods remains a **wide open** and yet important problem both in theory and practice.
>
> (See **Follow-up Response - Part 4**)

---

> ### Author Response · Authors · 2025-12-04
> **Follow-up Response - Part 4**
>
> Our work is centered around overcoming the above technical challenges. Our key contributions are summarized as follows:
>
> **1) A New Smoothed WC-Scalarization Approach for Pareto Front Exploration with DNN-Based Actor-Critic MORL:** To address the non-smoothness of the standard WC-scalarization, we propose a new smoothed-weighted Chebyshev (WC) scalarization technique to enable the use of policy gradients with WC-scalarization, while preserving the salient Pareto front exploration capability of the original WC-scalarization. Our proposed smoothed-WC-scalarization yields well-defined policy gradients that facilitate our subsequent actor-critic MORL algorithmic design.
>
> **2) A Judiciously Designed DNN-Based Actor-Critic Algorithms:** Through the bridge of our proposed smoothed WC-scalarization, we develop a **new** DNN-based actor-critic algorithmic framework with a **new** $Q$-function-based actor-critic design that significantly lowers the computational costs.
>
> **3) New Finite-Time Global Pareto-Optimality Convergence Rate Results for DNN-Based MORL:** With the techniques and insights from the previous two contributions, we further show that our proposed DNN-based actor-critic algorithm is the **first MORL method** that enjoys **global Pareto-optimality convergence** with an **$\mathcal{O}(1/T)$ finite-time convergence rate**, where $T$ denotes the total number of iterations. Moreover, our method guarantees systematic exploration of the entire Pareto front.
>
> **4) Our Theoretical Analysis Provides New Insights of Independent Interest:** In our theoretical analysis, we demonstrate (i) how performance difference lemma (Lemma 6.1 in [1]) can be made compatible with the MORL framework, (ii) why our algorithmic design effectively bridges the sub-optimality gap by computing a **new update direction** $d$, and (iii) the impacts of DNNs' width and length in the analysis and how to analyze them. These new ideas and techniques used in our work could be of independent interest to the general fields of multi-objective optimization (MOO) and reinforcement learning (RL).
>
> In addition to these theoretical contributions, our numerical experiments demonstrate that our approach outperforms other baselines and validates the correctness of our theories.
>
> In summary, we studied how to achieve systematic **Pareto front exploration** for **deep-neural actor-critic multi-objective reinforcement learning** (MORL) with **theoretical finite-time global Pareto-optimality convergence rate guarantee**, which is an open and important problem in both theory and practice. To address the technical challenges in solving this problem, we have developed a series of new algorithmic designs along with their theoretical performance guarantees, all of which are **new** in the literature and advance the state of the art of MORL research.
>
>
> ---
>
> [1] Approximately optimal approximate reinforcement learning. ICML, 2002.

---

### Official Review · Reviewer_bj58 · 2025-10-27

**Soundness:** 3
**Presentation:** 3
**Contribution:** 3
**Rating:** 6
**Confidence:** 3

**Summary:**

This paper develops a deep neural network-based actor-critic algorithm for multi-objective reinforcement learning that achieves global convergence to Pareto-optimal policies with a convergence rate of O(1/T). The authors use a smoothed weighted-Chebyshev scalarization technique to convert the multi-objective problem into a scalar-valued optimization problem, enabling systematic exploration of the entire Pareto front. They provide theoretical convergence guarantees and demonstrate the algorithm's effectiveness through experiments on recommendation systems and robotic simulations.

**Strengths:**

The paper makes good theoretical contributions by establishing the finite-time global convergence guarantees for DNN-based actor-critic methods in multi-objective reinforcement learning, addressing a notable gap between empirical practice and theory. The use of smoothed weighted-Chebyshev scalarization is well-motivated, enabling both global optimality and systematic Pareto front exploration through a principled mathematical framework. The theoretical analysis is rigorous, extending the performance difference lemma to the MORL setting and carefully handling the complexities introduced by neural network approximation. The paper is generally well-structured, with clear problem formulation and logical progression from motivation to algorithm design to convergence analysis. The experimental validation on recommendation systems and robotic simulation demonstrates practical applicability.

**Weaknesses:**

The paper has several limitations. The literature review is narrow, focusing primarily on scalarization methods while omitting important MORL approaches such as geometric analysis of the Pareto front, evolutionary methods, hypervolume-based techniques, and other non-scalarization paradigms, which limits proper contextualization of the contribution. The theoretical results rely on restrictive assumptions (particularly Assumptions 3-6) whose practical validity remains unverified, and the sample complexity suggests poor scalability to many-objective problems without empirical investigation of this limitation. The experimental evaluation is insufficient— the paper lacks comparison with recent DNN-based MORL methods beyond PDPG, making it difficult to assess whether the theoretical advantages translate to practical performance gains over existing approaches. Additionally, important practical considerations such as hyperparameter sensitivity, computational cost analysis, and guidance for parameter selection are largely absent, limiting the work's applicability to real-world problems.

**Questions:**

no

---

> ### Author Response · Authors · 2025-11-25
> **Author Response - Part 1**
>
> > **Comment 1:** The literature review is narrow, focusing primarily on scalarization methods while omitting important MORL approaches such as geometric analysis of the Pareto front, evolutionary methods, hypervolume-based techniques, and other non-scalarization paradigms, which limits proper contextualization of the contribution.
>
> **Response:** Thanks for the comment. We agree with the reviewer that we should do a more comprehensive literature review on MORL and include other MORL approaches suggested by the reviewer. In the original submission, we focused more on MORL methods that have theoretical performance guarantees. Following the reviewer's suggestion, in addition to the related works discussed in Section 2, we present additional related work review of MORL in **Appendix A**. We primarily focus on geometric and Pareto-based methods, evolutionary approaches, and also check the works that provide practical guidance for MORL. We thank the reviewer again for this suggestion.

---

> ### Author Response · Authors · 2025-11-25
> **Author Response - Part 2**
>
> > **Comment 2:** The theoretical results rely on restrictive assumptions (particularly Assumptions 3-6) whose practical validity remains unverified, and the sample complexity suggests poor scalability to many-objective problems without empirical investigation of this limitation.
>
> **Response:** Thanks for your comment. We would like to clarify why our Assumptions 3-6 are not very restrictive:
>
> **1) Assumptions 3-5 in this paper are standard and have been widely adopted in the literature.** We argue that Assumptions 3–5 are standard and have been widely used in the RL literature (e.g., [1-6]).
>
> **2) Assumption 6 is not restrictive and can be verified.** We would like to clarify that Assumption 6 is not used for guaranteeing the existence scalarization. Rather, Assumption 6 is used to lower bound the gradient norm of our proposed $\mu$-smoothed-approximated WC-scalarized objective function $G_\mu(\cdot)$.
>
> Moreover, Assumption 6 is not restrictive and can be easily satisfied in many MORL problems in practice. Specifically, since the dimension of DNN model parameters $\theta$ is typically **much larger than** the number of objectives $m$, we only require the smallest **singular value** of a "tall and skinny" matrix to be bounded away from 0. In other words, this only requires the following matrix:
> $$
> H(\theta | p) := \left[\begin{align*}
> & \hspace{1em}\vdots & & \hspace{1em}\vdots \\\\
> & \nabla_\theta \big(p_1(J_1^{\text{ub}} - J_1(\theta))\big) & \cdots\hspace{2em} & \nabla_\theta \big(p_m(J_m^{\text{ub}} - J_m(\theta))\big) \\\\
> & \hspace{1em}\vdots & & \hspace{1em}\vdots
> \end{align*} \right] \in \mathbb{R}^{n\times m}
> $$
> to have **$m$ linearly independent columns**, where $n:=\mathrm{dim}(\theta)$. Since $n \gg m$ and the $m$ tasks' objective and datasets are usually **independent**, the $H(\theta|p)$ matrix is **full column-rank** typically, which implies that **Assumption 6 holds**. Therefore, Assumption 6 is typically safe for MORL problems.
>
> In addition, we also numerically verify the validity of Assumption 6. Specifically, in the robotic simulation experiment, we store the upper bound of the score function's $\ell_2$-norm and the smallest singular value of the $H$ matrix at every time step, and obtain the following numerical results from the robotic simulation dataset:
>
> * The **smallest singular value** of the $H$ matrix is: **0.593**, which clearly **satisfies** Assumption 6.
> * The upper bound of the score function's $\ell_2$-norm is: **$1.51\times 10^2$**.
>
> We believe that these numerical results further validate our clarification, namely, that the assumptions used in this paper are both reasonable and unrestrictive.
>
> **3) The superlinear $\mathcal{O}(m^{\frac{3}{2}})$-scaling in the number of objectives is a typo. The correct scaling should be $\mathcal{O}(m^{\frac{1}{2}})$, which is sublinear.** Specifically, at each inner loop iteration, the action sampling occurs once as stated in Line 6 of Algorithm 1. Correspondingly, the sample complexity should be $T(M+K)$ instead of $T(mM+mK)$. Thus, choosing $K=\Omega(m^{\frac{1}{2}})$ as in Corollary 1 yields an $\mathcal{O}(m^{\frac{1}{2}})$-scaling with respect to the number of objectives $m$. This sublinear $\mathcal{O}(m^{\frac{1}{2}})$-scaling, coupled with the fact that the number of objectives $m$ is typically not large (rarely more than $100$ for applications in practice), implies that the scalability to many-objective problems is not a major concern. We have fixed this typo in our revision.
>
> ---
>
> [1] Theoretical study of conflict-avoidant multi-objective reinforcement learning. IEEE Transactions on Information Theory, 2025.
>
> [2] Finite-time convergence and sample complexity of multi-agent actor-critic reinforcement learning with average reward. ICLR, 2022.
>
> [3] An improved analysis of (variance-reduced) policy gradient and natural policy gradient methods. NeurIPS, 2020.
>
> [4] On the global optimum convergence of momentum-based policy gradient. AISTATS, 2022.
>
> [5] Closing the gap: Achieving global convergence (last iterate) of actor-critic under markovian sampling with neural network parametrization. ICML, 2024.
>
> [6] Finite-time convergence and sample complexity of actor-critic multi-objective reinforcement learning. ICML, 2024.

---

> ### Author Response · Authors · 2025-11-25
> **Author Response - Part 3**
>
> > **Comment 3:** The experimental evaluation is insufficient— the paper lacks comparison with recent DNN-based MORL methods beyond PDPG, making it difficult to assess whether the theoretical advantages translate to practical performance gains over existing approaches.
>
> **Response:** Thanks for your comments. As mentioned in our response to the reviewer's Comment 1, we agree that there are many DNN-based MORL methods in the literature, and we have followed the reviewer's suggestion to include those related works in our revision. Regarding the experimental evaluations, as mentioned in **Appendix A**, many of these MORL approaches in the related work section are designed for different problem settings, and hence not directly comparable.
>
> Also, we would like to clarify that the baseline PDPG, as well as the ablation study on the linear scalarization-based actor-critic, are both DNN-based methods, which are all directly comparable. This is the reason that we chose them as baselines in our original submission.
>
> Following the reviewer's suggestion, in this rebuttal period, we are trying to conduct additional numerical experiments with more DNN-based methods [1,2]. Due to the amount of extra experiments, we are still in the process working on these additional experiments, and we will update the numerical results as soon as these experiments are completed.
>
> ---
>
> [1] Two-Stage Constrained Actor-Critic for Short Video Recommendation. ACM web conference 2023.
>
> [2] Prediction-Guided Multi-Objective Reinforcement Learning for Continuous Robot Control. ICML, 2020.

---

> ### Author Response · Authors · 2025-11-25
> **Author Response - Part 4**
>
> > **Comment 4:** Additionally, important practical considerations such as hyperparameter sensitivity, computational cost analysis, and guidance for parameter selection are largely absent, limiting the work's applicability to real-world problems.
>
> **Response:** Thanks for your comment. We would like to respond to multiple distinct aspects in this comment point-by-point as follows:
>
> **1) Hyperparameter sensitivity:** The sensitivity of the two most important hyperparameters, the actor batch-size $M$ and the number of critic steps $K$, on the performance of our algorithm can be **theoretically understood** in Theorem 1. In particular, the overall convergence error bound (i) decreases at a rate of $\frac{1}{K}$ as the number of critic update steps iterations $K$ increases (corresponding to the critic component), and (ii) decreases at a rate of $\frac{1}{M^{\frac{1}{2}}}$ as the batch-size $M$ of the Markovian-sampling increases.
>
> **2) Computation costs:** We conducted all of our numerical experiments on a single H100, with the running time for each seed taking less than 2 hours. This shows that the computational cost is not a bottleneck for our approach.
>
> **3) Parameter selection:** The parameter selection can also be theoretically guided by Theorem 1. Specifically, the finite-time convergence error bound shrinks at a fast rate $\mathcal{O}(D^{-4})$ as the depth of the DNN $D$ increases, while the change in the width of the DNNs $w$ only lead to a slow drop of the convergence error bound at a rate of $\mathcal{O}(w^{-\frac{1}{24}})$, which is close to a constant. These theoretical results, coupled with a provided convergence error tolerance bound, can guide the DNN architecture and size parameter selection in practice. The above discussions on parameter selection are summarized in our Remark 6 (cf. Lines 423-426).

---

### Official Review · Reviewer_LnkA · 2025-10-28

**Soundness:** 1
**Presentation:** 2
**Contribution:** 1
**Rating:** 2
**Confidence:** 4

**Summary:**

This paper examines the problem of Multi-Objective Reinforcement Learning (MORL). It proposes an algorithm in the family of deep neural actor-critic methods to solve it. This algorithm relies on a smoothed Chebyshev scalarization to transform the MORL problem into scalar optimization. The main contribution of the paper is claimed to be that this algorithm enjoys provable convergence to the global optimum, while also providing a systematic exploration of the Pareto front as the scalarization free parameter is varied. Experiments on two MORL setups are claimed to demonstrate the practicality of the method.

**Strengths:**

The idea of a smoothed Chebyshev scalarization can prove useful when exploring highly concave Pareto Fronts in MORL.

The exposition of background for MORL is also comprehensive and precise.

Finally, the topic of study is highly relevant. The authors correctly identify that provable guarantees for DNN-based methods in MORL are lacking in the literature, at least to my knowledge.

**Weaknesses:**

Generally, one has to be suspicious of any claim of provable global convergence of deep neural networks. This problem is, to my knowledge, unsolved, even without the added complexity of Reinforcement Learning. For example, one popular object of study in the theory literature are "deep linear networks," which are DNNs without activation layers. This major simplification speaks to the difficulty of the original challenge.

# Problem with Assumption 4
Unfortunately, in my understanding, this paper fails to address the DNN global convergence adequately. In particular, I believe that the Assumption 4 that the paper introduces never holds for the class of policies that the paper considers. On L220-224, the paper introduces DNNs _with ReLu activation_. This activation function is scale-invariant, i.e. $ReLu(\alpha x)=\alpha ReLu(x)$ for $\alpha>0$. This introduces a problem that is well-known in the literature, namely the fact that the parameterization of DNNs with ReLu is redundant. Indeed, one can scale the weight and bias on layer $b-1$ by an arbitrary factor $\alpha>0$ and then scale the layer $b$ by $1/\alpha$. This results in an identical mapping, in our case an identical policy $\pi_\theta(a|s)$. What this means is that there is a vector $\mathbf{v}$ such that $\nabla_\theta \log \pi_\theta(a|s)$ is orthogonal to $\mathbf{v}$ _for any state $s$ and action $a$_. This in turn implies that $\mathbf{v}$ is a null-vector of $\mathbb{E}[\nabla_\theta \log \pi_\theta(a|s)\nabla^\intercal_\theta \log \pi_\theta(a|s)]$, which makes Assumption 4 unattainable. The paper cites multiple prior works to back up the reasonability of the assumption. These papers, however, are much more careful when introducing it:
- Liu et al, 2022. They introduce this assumption, but in the paper they only discuss that it holds _for linear policies_ with Gaussian noise. In Appendix B.2, which is also referred to by several other authors, they discuss the caveats of this assumption, mentioning that it _may_ hold for some non-linear classes of policies. They also mention that this is a _minimal_ requirement for proving convergence of their algorithm, which may close off this avenue for proving global convergence of DNNs with ReLu. I am less certain of the last part because I'm not sure how the algorithm from this paper relates to that of Liu et al.
- Agarwal et al, 2021. On cursory scan, they seem to only discuss assumption 5 from this paper, and not 4. Instead of 4, they use a different relative conditioning assumption (Assumption 6.5 in their paper).
- Ding et al, 2021. They also discuss this assumption in Appendix 8 and refer to Liu et al. They purposefully stay abstract and do not rely on specific DNN policies, only discussing deep learning a bit in the intro.

I believe that this is the reason why prior works in _provable_ single- and multi-objective RL could only prove something for much simpler classes of policies.

More generally, for a paper whose main contribution is an algorithm with provable convergence, I would expect to see some attempts to verify whether the assumptions hold in practice. One can, for example, estimate the Fischer information matrix by collecting rollouts on the practical environments, and computing the empirical expected value matrix. Then, one could check whether its lowest eigenvalue is non-zero and how large is $\sigma$. This paper provides no such analysis.

If I am not wrong about this assumption, in my opinion this issue is sufficient on its own to warrant a rejection.

## Problems with the experimental setup

The experimental validation of the algorithm, to which the paper dedicates less than a page, is also rather limited. For the first evaluation Sec. 6.1), the authors compare the values of the individual rewards to some baselines. It remains unclear from the main paper and from the (short) Appendix B.1 what was the preference vector that was used for comparison, and even whether this vector is set to the same value for the baselines. The authors also perform an evaluation to see if the Pareto front is adequately covered. From L454-457, my understanding is that they run their algorithm as a _single-objective_ RL problem by maxing out the preference on individual objectives. This is inadequate as a method for evaluating MORL. An overview work on MORL that is often used as a reference in the field [1] dedicates the entire Section 8 to evaluation methods. These include metrics such as hypervolume, the $\epsilon$-metric, utility-based, etc. None of these are reported in this paper.

For the second evaluation environment (Sec. 6.2), the paper uses the Walker environment from Gymnasium. Here, the paper does provide a plot of the Pareto front, but does not show how the baselines perform.

[1] Hayes, Conor F., et al. "A practical guide to multi-objective reinforcement learning and planning." arXiv preprint arXiv:2103.09568 (2021).

## Typos & other minor issues
- Algorithm 1, L7: I assume you want to sample the action and get $\nabla\log\pi$ outside of the inner loop

l170: have->has
l220: a->an
l259: "that," comma not needed
l375: should say "positive semi-definite order"
l461: mojoco->mujoco

**Questions:**

For the second evaluation environment (Sec. 6.2), the paper uses the Walker environment from Gymnasium. The single-objective imlementation of Gymnasium is cited, but a multi-objective environment is used. Do you actually use the MO-Gymnasium [2] or re-implement the multi-objective environment? I suspect it's the former, in which case it would be better to cite the actual environment.

[2] Felten, Florian, et al. "A toolkit for reliable benchmarking and research in multi-objective reinforcement learning." Advances in Neural Information Processing Systems 36 (2023): 23671-23700.

Also, what are the types of distribution parameterizations that the paper uses for the policy $\pi(a|s)$?

---

> ### Author Response · Authors · 2025-11-22
> **Author Response - Part 1**
>
> > **Comment 1:** Problem with Assumption 4.
>
> **Response:** We sincerely appreciate the reviewer's insightful comment. The reviewer's comment regarding Assumption 4 and ReLU activation is *correct,* and we indeed overlooked the null space issue caused by the ReLU activation function. As the reviewer pointed out, due to the scale-invariant property of the ReLU function, the parameterization becomes redundant. As a result, we cannot directly assume that the Fisher matrix is semi-positive definite. Therefore, we agree that for ReLU-based DNNs, Assumption 4 does not hold.
>
> Fortunately, this problem can be fixed and our main results are not affected. We summarize how to address this issue as follows:
>
> **1) Universal approximation still holds with other activation functions:** Note that the **rationale** of using ReLU in our original submission is to leverage its **universal approximation** result in Lemma 2. Therefore, the use of ReLU is *not necessary* in our work. Specifically, **as long as we could replace ReLU by other non-scale-invariant activation functions that enjoy universal approximation, our main theoretical results will not be affected.**
>
> Upon further studies in this rebuttal period, we conclude that the clash between Assumption 4 and ReLU can be resolved by using other activation functions, which can still maintain the validity of the paper. Specifically, as outlined in [1,2], the universal approximation capability of multi-layer perceptrons (MLPs) with various activation functions can be summarized as follows:
>
> * ReLU-based MLPs with depth $D$ and width $w$ can approximate any Lipschitz continuous function with a vanishing error rate of $\widetilde{\mathcal{O}}(w^{-2/d}D^{-2/d})$. This is exactly what we showed in our original **Lemma 2**.
> * MLPs with activation functions （e.g., **Sigmoid**) achieve the **same** order of universal approximation error as ReLU-based MLPs. Specifically, the set of MLPs with width no greater than $3w$ and depth no greater than $2D$, i.e., $\mathcal{NN}(3w, 2D; Sigmoid)$ is dense in $\mathcal{NN}(w, D; ReLU)$. This implies that, for the Lipschitz continuous function, **the universal approximation error of a sigmoid-based $(3w,2D)$-MLPs also vanishes at the same rate of $\widetilde{\mathcal{O}}(w^{-2/d}D^{-2/d})$**.
> * Similar results also hold for other activation functions such as **tanh**, **Arctan**, and **SoftSign**.
>
> **2) Salvaging Assumption 4 by using non-scale-variant activation functions with universal approximation effects:** Based on 1), we can restate the DNN parameterization in our paper by replacing the **ReLU** activation function with the **Sigmoid** activation function. The **Sigmoid** activation function does **not** suffer from the scale-invariant issue, indicating that the Fisher matrix can satisfy the semi-positive definite property. Thus, **our theoretical analysis and the main convergence result continue to hold after replacing ReLU by Sigmoid or other non-scale-invariant activation functions capable of universal approximation**.
>
> Again, we thank the reviewer for pointing out this technical issue, which helps improve this paper. We have already addressed this problem in the updated revision (cf. Lines 220-242).
>
> ---
>
> [1] Optimal approximation rate of relu networks in terms of width and depth. 2022.
>
> [2] Deep network approximation: Beyond relu to diverse activation functions. JMLR, 2024.

---

> > ### Comment · Reviewer_LnkA · 2025-11-26
> > **Switching activation functions is unsatisfactory**
> >
> > I appreciate that the authors acknowledge the problem that I pointed out with Assumption 4. It is true that universal approximation holds with sigmoid or similar activations as well. It is also true that these other activations are not scale-invariant, so this particular avenue for discovering null vectors of the Fisher information matrix is then closed. However, I don't think that changing the activation fixes the underlying issue. Fundamentally, the problem that gives rise to these null vectors is that the mapping from the space of weights of the MLP into the space of functions represented by the MLP is often not injective. For example, changing the incoming weights connected to a "dead" neuron (whose outgoing weights are zero) is not going to influence the final function represented by the MLP. I suspect that there are more invariances like this that can be provided for any given activation function. Since the paper does not provide an experimental evaluation to establish the value of $\sigma$ in Assumption 4, I remain skeptical that the problem can be resolved.

---

> ### Author Response · Authors · 2025-11-22
> **Author Response - Part 2**
>
> > **Comment 2:** Problems with the experimental setup.
> > The experimental validation of the algorithm, to which the paper dedicates less than a page, is also rather limited. For the first evaluation Sec. 6.1), the authors compare the values of the individual rewards to some baselines. It remains unclear from the main paper and from the (short) Appendix B.1 what was the preference vector that was used for comparison, and even whether this vector is set to the same value for the baselines. The authors also perform an evaluation to see if the Pareto front is adequately covered. From L454-457, my understanding is that they run their algorithm as a single-objective RL problem by maxing out the preference on individual objectives. This is inadequate as a method for evaluating MORL. An overview work on MORL that is often used as a reference in the field [1] dedicates the entire Section 8 to evaluation methods. These include metrics such as hypervolume, the $\epsilon$-metric, utility-based, etc. None of these are reported in this paper.
> >For the second evaluation environment (Sec. 6.2), the paper uses the Walker environment from Gymnasium. Here, the paper does provide a plot of the Pareto front, but does not show how the baselines perform.
>
> **Response:** Thanks for the comment. Due to the length of this comment, we would like to clarify our experimental setups and results point-by-point as follows:
>
> **1)** We agree with the reviewer that the description of the recommendation system experiment may not be clear enough. In this rebuttal period, we have added a detailed explanation in our revised **Appendix C.1**. Two key points are summarized as follows:
>
> * For Table 2, we note that some algorithms are preference-free, while MOCHA and our algorithm are preference-based. Therefore, we select a **single** preference vector for a fair comparison. To balance the performances across all objectives, especially given the highly biased dataset, we select $p = \frac{\lambda}{\|\lambda\|}$, where $\lambda = [10.0, 1.0, 1.0, 1.0, 0.01, 0.1, 10.0]^\top$ as the preference vector for MOCHA and our algorithm in Table 1.
> * For Figure 2, we do **not** select the one-hot preference vectors. Instead, the chosen vectors such as $P_{\text{click}}$ are listed in Lines 1167-1175 (Appendix C.1) of our revision. As a result, the corresponding problems still retain their multi-objective nature, and our results indeed demonstrate the systematic Pareto exploration capability of our algorithm. Besides, the "reference point" is still based on the preference vector selected for Table 2, i.e., $p = \frac{\lambda}{||\lambda||}$ where $\lambda = [10.0, 1.0, 1.0, 1.0, 0.01, 0.1, 10.0]^\top$.
>
> **2)** We also agree with the reviewer that the using more systematic metrics (e.g., Hypervolume and $\epsilon$-metric) can enhance our work. In this rebuttal period, we have added the **new** experimental results using the Hypervolume metric in our revision (cf. Lines 1177-1208 in our revision).
>
> **3)** During the rebuttal period, we also added new results for the robotic simulation experiment in **Section 6.2**. Since the recommendation system part has already provided evidence for the baseline comparison, in this robotic experiment, we compare our algorithm with a linear scalarization-based actor-critic method to verify the efficacy of the WC technique. These new results further validate the superior performance of our approach.

---

> > ### Comment · Reviewer_LnkA · 2025-11-26
> > **Extra experimental details**
> >
> > I appreciate that the authors added the details for the experimental evaluation, they do provide helpful context. However, I maintain my opinion that the experimental evaluation is insufficient.
> >
> > > For Table 2, we note that some algorithms are preference-free, while MOCHA and our algorithm are preference-based. Therefore, we select a single preference vector for a fair comparison.
> >
> > Do I understand correctly that by "preference-free" you mean that the algorithm discovers the entire Pareto front in one run, while a "preference-based" algorithm is rerun for each preference vector? In that case, it would make sense to me to select baselines appropriately (preference-based against preference-based) and compare the Pareto fronts after multiple runs with varying preference vectors.
> >
> > > For Figure 2, we do not select the one-hot preference vectors. Instead, the chosen vectors such as $P_{\text{click}}$ are listed in Lines 1167-1175 (Appendix C.1) of our revision. As a result, the corresponding problems still retain their multi-objective nature, and our results indeed demonstrate the systematic Pareto exploration capability of our algorithm.
> >
> > Yes, but from this table, the preference vectors are indeed skewed heavily to individual rewards, so the problems are almost single objective. I do not see that this experiment provides evidence for "systematic Pareto exploration." It is indeed true from Fig. 2 that when preference to one objective is set higher, this objective gets optimized more. However, it is not clear why this implies that the entire Pareto front is covered.
> >
> > > 2) We also agree with the reviewer that the using more systematic metrics (e.g., Hypervolume and $\epsilon$-metric) can enhance our work. In this rebuttal period, we have added the new experimental results using the Hypervolume metric in our revision (cf. Lines 1177-1208 in our revision).
> >
> > The added hypervolume discussion on L1185-1187 reads to me like a misunderstanding about how the metric is supposed to work. It makes little sense to compute it using only one preference vector, which is what the authors seem to be doing. Instead, the metric should be computed from a Pareto front that is constructed from many points, eg obtained by rerunning the optimization with many preference vectors.
> >
> > Unfortunately, I don't think the theoretical and experimental issues with this paper are fixable within the rebuttal period, and I maintain my recommendation for rejection.

---

> ### Author Response · Authors · 2025-11-22
> **Author Response - Part 3**
>
> > **Comment 3:** For the second evaluation environment (Sec. 6.2), the paper uses the Walker environment from Gymnasium. The single-objective imlementation of Gymnasium is cited, but a multi-objective environment is used. Do you actually use the MO-Gymnasium or re-implement the multi-objective environment? I suspect it's the former, in which case it would be better to cite the actual environment.
>
> **Response:** Thanks for your question and pointing out the citation issue. We confirm that we have used a **multi-objective** experimental environment, i.e., `mo-walker2d-v5`, and the correct citation should be the one mentioned by the reviewer. We appreciate the reviewer’s careful checking, and we have already corrected this issue in our revision.
>
> ---
>
> > **Comment 4:** What are the types of distribution parameterizations that the paper uses for the policy $\pi(a|s)$?
>
> **Response:** Thank you for your question. When implementing the robotic simulation experiments in Section 6.2, we apply the Gaussian distribution to represent the policy. We note that this Gaussian representation has also been widely used in the literature [1-3].
>
> [1] Proximal Policy Optimization Algorithms. ArXiv, 1707.06347.
>
> [2] Soft actor-critic: Off-policy maximum entropy deep reinforcement learning with a stochastic actor. ICML, 2018.
>
> [3] The definitive guide to policy gradients in deep reinforcement learning: Theory, algorithms and implementations. ArXiv, 2401.13662.
>
> ---
>
> > **Comment 5:** Typos and other minor issues.
>
> **Response:** Thankks for catching these typos and minor issues. We have already fixed them in this revision.

---

> ### Author Response · Authors · 2025-12-03
> **Follow-up Response - Part 1**
>
> > **Comment:** I appreciate that the authors acknowledge the problem that I pointed out with Assumption 4. It is true that universal approximation holds with sigmoid or similar activations as well. It is also true that these other activations are not scale-invariant, so this particular avenue for discovering null vectors of the Fisher information matrix is then closed. However, I don't think that changing the activation fixes the underlying issue. Fundamentally, the problem that gives rise to these null vectors is that the mapping from the space of weights of the MLP into the space of functions represented by the MLP is often not injective. For example, changing the incoming weights connected to a "dead" neuron (whose outgoing weights are zero) is not going to influence the final function represented by the MLP. I suspect that there are more invariances like this that can be provided for any given activation function. Since the paper does not provide an experimental evaluation to establish the value of $\sigma$ in Assumption 4, I remain skeptical that the problem can be resolved.
>
>
> **Response:** We sincerely thank the reviewer for pointing out the conflict between Assumption 4 and our use of ReLU-based MLPs in the original Lemma 2, which arises from ReLU’s "scale-invariant" property. We acknowledge that this was indeed an oversight. Previously, we have shown that this conflict can be resolved by replacing the activation function with Sigmoid, which avoids the scale-invariance issue while retaining comparable universal approximation capability. We are delighted to see that the reviewer has agreed the scale-invariant avenue to null space has been closed.
>
> Regarding the further question on the potential existence of null vectors in MLPs (due to MLPs' non-injectivity) that could still violate the Fisher non-degeneracy (FND) condition in Assumption 4, our response is summarized as follows:
>
> * **Assumption 4 Can Be Relaxed by Slightly Modifying Assumption 5:** We agree with the reviewer that the injective property does not hold for MLPs in general and null vectors could exist in MLPs. However, besides the anecdotal cases (e.g., "dead neurons"), the fundamental reason of the existence of such null vectors arises from potential "dimension reduction" between two consecutive layers in an MLP. Specifically, if there is any "short and fat" weight matrix between two consecutive layers in an MLP (i.e., projecting from a higher dimensional latent space at one layer onto a lower dimensional latent space at the next layer), a non-empty null space will exist.
>
>   To resolve this fundamental issue, our solution is to **relax** the FND requirement in Assumption 4 and slightly modify Assumption 5 by imposing a small perturbation requirement (cf. our new Assumption 5). Specifically, we note that the main purpose of assuming FND in Assumption 4 is to ensure that the norm of Fisher matrix $\\|F(\theta)\\|$ can be lower bounded away from zero by some $\sigma>0$, which facilitates a necessary step in many related works in the literature to achieve global optimality (see, e.g., Sec. 11 in [1]). With the newly added $\sigma$-perturbation in Assumption 5, we can directly guarantee the $\\|F(\theta)^{-1}\\| \le \frac{1}{\sigma}$ instead. By using the modified Assumption 5, our proofs and results will continue to hold, while avoiding the use of FND in Assumption 4 and any potential issues caused by MLPs' non-injectivity.
>
> * **FND in Assumption 4 Holds for MLPs with Non-decreasing Widths:** Although FND in the original Assumption 4 does not hold for MLPs in general, it is worth noting that FND can be nonetheless guaranteed by MLPs with non-decreasing widths (i.e., the width of one layer is never wider than its next layer). Also, any weight matrix between two consecutive layers should be of full column rank. Under these structural requirements, it can be guaranteed that the Sigmoid-based MLPs do have an empty null space, which further implies FND in Assumption 4. We also note that the above "non-decreasing width" and "full column-rank" MLP structural requirements can be viewed as a natural **extension** of the "full column-rank" condition for linear policies (e.g., [2]) to **nonlinear neural MLP policies**.
>
> ---
>
> [1] On the Global Optimum Convergence of Momentum-based Policy Gradient. AISTATS, 2022.
>
> [2] An Improved Analysis of (Variance-Reduced) Policy Gradient and Natural Policy Gradient Methods. NeurIPS, 2020.

---

> ### Author Response · Authors · 2025-12-03
> **Follow-up Response - Part 2**
>
> > **Comment:** Do I understand correctly that by "preference-free" you mean that the algorithm discovers the entire Pareto front in one run, while a "preference-based" algorithm is rerun for each preference vector? In that case, it would make sense to me to select baselines appropriately (preference-based against preference-based) and compare the Pareto fronts after multiple runs with varying preference vectors.
>
> **Response:** Thanks for your follow-up comment. We would like to further clarify as follows:
>
> **1) Misunderstanding of Preference-free Methods:** In multi-objective optimization (MOO) and multi-objective reinforcement learning (MORL) problems, the term "preference-free methods" means **no**  preference information is used as input. In general, preference-free methods **cannot** explore the Pareto front. Rather, these methods typically converge (if convergence happens at all) to an arbitrary Pareto solution. One prime example of preference-free method is the classical multi-gradient descent algorithm (MGDA), which guarantees the convergence to a Pareto-stationary point but lacks the control of which Pareto-stationary point to converge to.
>
> In contrast, preference-based methods leverage preference information to guide the convergence toward different points across the Pareto front, which holds the potential to enable systematic Pareto front exploration (but Pareto front exploration remains challenging even with the use of preference information).
>
> Our DNN-based actor-critic algorithm for MORL is a preference-based method, and one of our main contributions is that it offers the capability to systematically explore the entire Pareto front.
>
> **2) Clarification of Comparison between Preference-free and Preference-based Methods.** Our purpose of comparing preference-based and preference-free methods is to show that, through the control of preference vector, our preference-based method can achieve better performance than preference-free methods. As mentioned earlier, preference-free methods do not use preference information as input and only converge to an arbitrary Pareto solution. Therefore, to ensure a fair comparison between preference-free and preference-based methods, we use a **single fixed preference vector** in our preference-based method to obtain only one Pareto-optimal solution, rather than allowing multiple preference vectors to explore a set of Pareto-optimal solutions. As shown in Table 2, under the control of a specific preference vector, our **single-preference-based** approach outperforms the preference-free baselines across the majority of the objectives.

---

> ### Author Response · Authors · 2025-12-03
> **Follow-up Response - Part 3**
>
> > **Comment:** Yes, but from this table, the preference vectors are indeed skewed heavily to individual rewards, so the problems are almost single objective. I do not see that this experiment provides evidence for "systematic Pareto exploration." It is indeed true from Fig. 2 that when preference to one objective is set higher, this objective gets optimized more. However, it is not clear why this implies that the entire Pareto front is covered.
>
> **Response:** Thanks for your comment. We are pleased to see the reviewer acknowledge we did not select the one-hot preference vectors. We would like to further explain our preference selection and the relation to Pareto exploration as follows:
>
> 1. We acknowledge that calling Figure 2 "systematic Pareto exploration" could be misleading. The purpose of using "heavily skewed" preference vectors in Figure 2 is to "explore the boundary" of the Pareto front. In other words, this preference vector selection facilitates an efficient Pareto front boundary discovery. To avoid the confusion with "Pareto front exploration with even coverage" we have changed the caption of Figure 2 as *"Pareto front boundary discovery"* in our revision (cf. Line 465).
> 2. On the other hand, the systematic Pareto exploration capability of our algorithm is validated in Figure 3, where we uniformly sample multiple preference vectors across the standard simplex and characterize the Pareto front based on these vectors. It can be seen from Figure 3 that, with these evenly distributed preference vectors, our algorithm effectively performs a systematic Pareto exploration.

---

> ### Author Response · Authors · 2025-12-03
> **Follow-up Response - Part 4**
>
> > **Comment:** The added hypervolume discussion on L1185-1187 reads to me like a misunderstanding about how the metric is supposed to work. It makes little sense to compute it using only one preference vector, which is what the authors seem to be doing. Instead, the metric should be computed from a Pareto front that is constructed from many points, eg obtained by rerunning the optimization with many preference vectors.
>
> **Response:** Thanks for your comment on the hypervolume (HV) metric. We agree with the reviewer on the definition of the HV metric, and we indeed have used a "simplified" version of HV metric. We would like to clarify the results with this simplified HV metric as follows:
>
> **1)** As discussed earlier, the preference-free methods **cannot** output multiple non-dominant points. Therefore, the HV metric of preference-free methods degenerates to the following "single point" form with respect to a reference point:
> $$\text{HV} = \Pi_{i=1}^{m} \|\text{Alg}_i - \text{Ref}_i\|,$$
> where $\text{Alg}_i$ denotes the output of the $i$-th objective for the preference-free algorithm, and $\text{Ref}_i$ denotes the corresponding component $i$ of the reference point.
>
> **2)** For the standard HV computation with a union of non-dominant solutions, we simplify it by using only one solution in Table 2 based on the following *rationale:* if the "box volume" obtained from one solution point under our approach is already larger the HV achieved by a preference-free baseline method, then the true HV of our method is **even larger** than the HV of the preference-free baselines, since the union of the "box volumes" of non-dominant solutions obtained from our method is non-decreasing as the size of the union increases. In addition, for the preference-based methods, we indeed compute the HVs using a union of non-dominant solutions for Figure 3 here (with reference point $(0, -5\times 10^2)$):
>
> |  | Ours | Baseline |
> | -------- | -------- | -------- |
> | HV($\uparrow$)     | **2.936**     | 2.213     |

---

### Official Review · Reviewer_ePuK · 2025-11-02

**Soundness:** 3
**Presentation:** 3
**Contribution:** 2
**Rating:** 6
**Confidence:** 3

**Summary:**

This paper studies the theoretical foundations of DNN–based actor–critic methods with weighted Chebyshev scalarization for MORL.  Experiments on recommendation and robotic control tasks are included to illustrate the theoretical results.

**Strengths:**

The paper presents a well-defined theoretical contribution, analyzing DNN-based actor–critic methods under the weighted Chebyshev (WC) scalarization framework.

The authors provide conditions under which a finite-time global convergence proof with an O(1/T) rate and accompanying sample complexity analysis is possible.

**Weaknesses:**

While the theoretical contribution is addressed, the use of weighted Chebyshev scalarization in MORL field itself is not a new idea. The scope of novelty is not sufficiently emphasized; the contribution could be better positioned relative to prior MORL work using similar scalarization techniques.

The main contribution is theoretical, and the algorithmic or methodological novelty beyond the convergence analysis is limited.

Fig. 4, or in fact the paper as a whole, does not contain any statistical information regarding reliability. The data shown here is a study of the parameter settings, not necessarily an ablation study. A a network with $D\le 4$ layers is usually not considered as a deep network, it's just a feed-forward network.

**Questions:**

Beyond theoretical guarantees, how might the proposed convergence analysis inform the design or improvement of practical MORL algorithms?

Could the authors clarify what aspect of their approach represents the main novelty compared with existing WC-based scalarization methods? Likewise, we could ask whether Assumption 6 is already guaranteeing that a scalarization exists that is sufficient to solve the problem, which doesn't apply in MORL in general.

The control cost is defined (only in the appendix) as a squared norm, but in Figure 3 it assumes also negative values. Also, in seems to be maximized rather than minimized (as stated in the text). Can you check this and provide the definitions completely and already when needed?

---

> ### Author Response · Authors · 2025-11-24
> **Author Response - Part 1**
>
> > **Comment 1:** While the theoretical contribution is addressed, the use of weighted Chebyshev scalarization in MORL field itself is not a new idea. The scope of novelty is not sufficiently emphasized; the contribution could be better positioned relative to prior MORL work using similar scalarization techniques.
>
> **Response:** Thanks for the comment. We agree that the weighted-Chebyshev (WC) scalarization technique itself is not a brand new idea. However, the **novelty** of our work lies in using the a **smoothed WC technique as a bridge to establish the first theoretical finite-time convergence rate guarantee for DNN-based actor-critic MORL approaches** in the literature. In other words, our focus in this work is on developing a **DNN-based actor-critic MORL** algorithm that is capable of systematic **Pareto front exploration** with **theoretical finite-time convergence rate**, rather than the WC technique itself. We summarize the key contributions and novelties as follows:
>
> **1) Our Smooth-WC-based Actor-Critic MORL Algorithmic Framework is Novel:** While the WC scalarization has been adopted in the literature of multi-objective optimization (MOO), the investigation of WC-based actor-critic MORL remains limited because WC's non-smooth "min-max" nature is not directly applicable for policy-gradient-based MORL methods. To overcome this key challenge, we propose a **new smoothed-WC-scalarization** to enable **well-defined policy gradient** computation and prove our proposed smoothed-WC-technique **still preserves the 1-to-1 mapping between smoothed WC-scalarized objective and the original Pareto front, hence facilitating systematic Pareto front exploration**. We note that all of these results are **new** in the literature. Moreover, our smoothed-WC-based actor-critic MORL approach enjoys **finite-time global convergence** for DNN-based nonlinear actor-critic MORL problems, which is also **new** in the literature.
>
> **2) Contribute to the Theoretical Foundation for DNN-based Actor-Critic MORL Algorithms:** Current state-of-the-art MORL methods are limited in the following key aspects:
>
> * Deep neural network (DNN)-based MORL approaches, though widely adopted in practice, are primarily **heuristic**, lacking theoretical justifications and guarantees for their performances.
> * Theoretical convergence and sample complexity results are only limited to actor-critic MORL methods based on **linear** function approximations.
>
> Our work addresses these limitations by overcoming the technical challenges arising from the combination of nonlinear function approximation and the non-smoothed WC-based systematic Pareto front exploration of MORL. We establish the first **finite-time global convergence** for DNN-based actor-critic MORL algorithms.
>
> **3) Our Theoretical Analysis Provides New Insights of Independent Interest:** In our theoretical analysis, we demonstrate (i) how performance difference lemma (Lemma 6.1 in [1]) can be compatible with the MORL framework, (ii) why our algorithmic design effectively bridges the sub-optimality gap by computing a novel update direction $d$, and (iii) the role of DNNs in the analysis and how they are handled. These new ideas and techniques used in this work could be of general interest to the general fields of multi-objective optimization and reinforcement learning.
>
> ---
>
> [1] Approximately optimal approximate reinforcement learning. ICML, 2002.

---

> ### Author Response · Authors · 2025-11-24
> **Author Response - Part 2**
>
> > **Comment 2:** The main contribution is theoretical, and the algorithmic or methodological novelty beyond the convergence analysis is limited.
>
> **Response:** Thanks for your comment. We would like to re-emphasize and clarify our algorithmic and methodological novelties as follows:
>
> **1) Our actor-critic algorithmic design considers a complex-structured problem in Eq. (2) that is challenging for policy gradient evaluation.** In the conventional actor-critic algorithms, the actor component typically follows the policy gradient theorem based on a general total reward function. However, in our WC-scalarized MORL setting, the actor component considers a smoothed-approximated-WC-scalarization of a reformulated MORL problem (polarity direction is changed from reward maximization to regret minimization to fit the standard WC-MOO framework). Therefore, the policy gradient direction corresponds to a complex-structured objective function $G_\mu$ in Eq. (2). As a result, the gradient direction of $G_{\mu}$ in Eq. (3) needs to include all of the $m$ objective functions' upper bounds and their first-order information, which significantly complicates the actor-critic design as seen in Algorithm 1. We note that such a special-structured actor-critic design is **new** in the literature.
>
> **2) New critic design based on $V(\cdot)$- rather than $Q(\cdot,\cdot)$-functions, necessitating new proofs.** Also due to the complex-structured smoothed-approximated-WC-scalarized objective $G_\mu$ in Eq. (2), one major challenge in our actor-critic MORL algorithm design arises from Line 13 of Algorithm 1, where not only the **gradients** but also the **objective functions themselves** are required to determine the descent direction. This is **unseen** in the single-objective RL literature. Such a specific requirement makes it **intractable** for algorithms that maintain $Q(\cdot,\cdot)$-value functions to scale in settings with **large or continuous action spaces**, as they require traversing the action space to compute the Q-values. To address this challenge, we deviate from conventional critic designs with nonlinear function approximation in the literature that are typically based on $Q(\cdot,\cdot)$-functions. Instead, we propose a **clean-slate** new critic design based on $V(\cdot)$-functions, which is major **novelty** that successfully avoids the computation tractability challenge due to the use of $Q(\cdot,\cdot)$-functions. However, this shift from $Q(\cdot,\cdot)$- to $V(\cdot)$-functions necessitate new proof and analysis techniques, which is another major **novelty** of our work. In this revision, we include the above discussions on actor-critic design challenges and our new critic design philosophy in **Remark 3**.
>
> **3) Non-MGDA-based actor-critic MORL algorithm design and its global convergence proofs are all new.** We note that most of the existing actor-critic MORL approaches with theoretical finite-time convergence rate guarantees have also adopted the multi-gradient descent algorithm (MGDA) design (e.g., [1]), which can be viewed as a natural extension of the classical gradient descent method in the multi-objective optimization paradigm. However, a key limitation of these existing MGDA-based actor-critic MORL algorithms is that they can at best guarantee the convergence to Pareto stationary solutions rather than Pareto-optimal solutions (i.e., local convergence rather than global convergence). To overcome this limitation, in this work, we propose a non-MGDA-based actor-critic MORL approach that solely exploits the special structure of the smoothed-approximated-WC-scalarized objective $G_\mu(\cdot)$, which is the **key** that establishes finite-time **global convergence guarantee** of our proposed actor-critic MORL approach to Pareto-optimal policies. The above discussions are further elaborated in our **Remark 4**.
>
> ---
>
> [1] Enabling pareto-stationarity exploration in multi-objective reinforcement learning: A multi-objective weighted-chebyshev actor-critic approach. ArXiv, 2507.21397.

---

> ### Author Response · Authors · 2025-11-24
> **Author Response - Part 3**
>
> > **Comment 3:** Fig. 4, or in fact the paper as a whole, does not contain any statistical information regarding reliability. The data shown here is a study of the parameter settings, not necessarily an ablation study. A a network with $D\le 4$ layers is usually not considered as a deep network, it's just a feed-forward network.
>
> **Response:** Thanks for your comment. Due to multiple distinct aspects in this comment, we would like to respond point-by-point as follows:
>
> **1) Statistical Information:** We agree with the reviewer that having statistical information regarding reliability can significantly enhance our numerical results. As the reviewer suggested, we have added the 95% confidence intervals in Fig. 5 of our revision.
>
> **2) Ablation Study:** We also agree with the reviewer that the use of the term "ablation study" is imprecise, and a more appropriate terminology is an investigation of parameter's impact or sensitivity. Meanwhile, it remains important to conduct ablation studies for our proposed DNN-based actor-critic MORL method. Thus, in this rebuttal period, we have added a new ablation study in our revision to demonstrate how WC-scalarization outperforms linear scalarization (i.e., the ablation study on the impact of WC-scalarization), and the results can be found in **Section 6.2** and **Appendix C.2** in our revision.
>
> **3) Not a Deep Network:** We agree that $D=4$ is not considered a deep neural network (DNN). We set $D=4$ in our original submission primarily due to computing resource limitation. Although it is challenging for us to conduct experiments with large depth, we note that another important conclusion in our theoretical result is that increasing the depth $D$ of the DNN improves the overall performance, which can be seen in the experimental results in our original submission. To further strengthen this result, during this rebuttal period, we have conducted new experiments by doubling the depth, i.e., $D=8$ in Fig. 5(a). We also point out that, for this robotic experiment, the action space has a relatively small dimension. As a result, relatively "shallow" DNNs are sufficient to achieve good performances.

---

> ### Author Response · Authors · 2025-11-24
> **Author Response - Part 4**
>
> > **Comment 4:** Beyond theoretical guarantees, how might the proposed convergence analysis inform the design or improvement of practical MORL algorithms?
>
> **Response:** Thanks for your comment. We provide the following insights regarding how our theoretical analysis inform the design or improve MORL in practice:
>
> **1) Our theoretical analysis reveals how DNNs' depth and width impact the performance and can guide the selection of DNNs' size.** As shown in Theorem 1, the finite-time convergence error bound shrinks at a fast rate $\mathcal{O}(D^{-4})$ as the depth of the DNN $D$ increases, while the change in the width of the DNNs $w$ only lead to a slow drop of the convergence error bound at a rate of $\mathcal{O}(w^{-\frac{1}{24}})$, which is close to a constant. These insights can guide the DNN architecture and size selection in practice.
>
> **2) Our theoretical analysis provides guidelines on how to choose hyperparameters.** In addition to the DNN size selection, our convergence analysis results also highlight how hyperparameters of the system (e.g., the number of objectives, the Markovian-sampling batch size, and the smooth-WC parameter influence the algorithm) affect the finite-time error convergence bound, which provide useful guidelines on choosing these hyperparameters in practice.

---

> ### Author Response · Authors · 2025-11-24
> **Author Response - Part 5**
>
> > **Comment 5:** Could the authors clarify what aspect of their approach represents the main novelty compared with existing WC-based scalarization methods? Likewise, we could ask whether Assumption 6 is already guaranteeing that a scalarization exists that is sufficient to solve the problem, which doesn't apply in MORL in general.
>
> **Response:** Thanks for your comments. We would like to answer the two distinct subquestions point-by-point as follows:
>
> **1) Aspects in our approach that are new compared to existing WC-scalarization methods:** Existing works in literature used the WC-scalarization technique (e.g., [1,2]) as regularization term in the objective coupled with MGDA to induce preference-following and systematic Pareto front exploration, rather than using WC-scalarization directly as the objective in actor-critic design for MORL. However, as mentioned earlier, a major limitation of such MGDA-based approaches is that they can only guarantee local Pareto stationarity convergence rather than global Pareto optimality guarantee. To overcome this limitation, a key **novelty** in this work is that we directly use WC-scalarization as the objective in our actor-critic design with nonlinear function approximation.
>
> However, due to WC's non-smooth "min-max" nature, it is not directly amenable for policy-gradient-based MORL methods. Another major **novelty** in this work is to abandon the standard WC-scarlization and instead work with a "**$\mu$-smoothed approximation**" of the WC-scalarization to facilitate well-defined policy gradients. However, this smoothed approximation approach raises another fundamental question: *"Could the $\mu$-smoothed approximation destroy the most salient feature of the original WC-scalarization: the 1-to-1 correspondence between the WC-objective and the Pareto front (key for Pareto-front exploration)?"* In this work, the third **novelty** is that we show such a 1-to-1 correspondence continue to hold between the $\mu$-smoothed-approximated WC-objective and the original Pareto front with sufficiently small $\mu$-values by leveraging latest advancements in MOO. All such results finally culminate the **finite-time global convergence** and **systematic Pareto front exploration** for DNN-based nonlinear actor-critic MORL problems, both of which are **new** in the literature.
>
>
> **2) "Assumption 6 guarantees the existence of scalarization but doesn't apply in MORL in general?"** Here, we would like to clarify that Assumption 6 is not used for guaranteeing the existence scalarization. Rather, Assumption 6 is used to lower bound the gradient norm of our proposed $\mu$-smoothed-approximated WC-scalarized objective function $G_\mu(\cdot)$.
>
> Moreover, Assumption 6 is not restrictive and can be easily satisfied in many MORL problems in practice. Specifically, since the dimension of DNN model parameters $\theta$ is typically **much larger than** the number of objectives $m$, we only require the smallest **singular value** of a "tall and skinny" matrix to be bounded away from 0. In other words, this only requires the following matrix:
> $$
> H(\theta | p) := \left[\begin{align*}
> & \hspace{1em}\vdots & & \hspace{1em}\vdots \\\\
> & \nabla_\theta \big(p_1(J_1^{\text{ub}} - J_1(\theta))\big) & \cdots\hspace{2em} & \nabla_\theta \big(p_m(J_m^{\text{ub}} - J_m(\theta))\big) \\\\
> & \hspace{1em}\vdots & & \hspace{1em}\vdots
> \end{align*} \right] \in \mathbb{R}^{n\times m}
> $$
> to have **$m$ linearly independent columns**, where $n:=\mathrm{dim}(\theta)$. Since $n \gg m$ and the $m$ tasks' objective and datasets are usually **independent**, the $H(\theta|p)$ matrix is **full column-rank** typically, which implies that **Assumption 6 holds**. Therefore, Assumption 6 is typically safe for MORL problems.
>
> In addition, we also numerically verify the validity of Assumption 6. Specifically, in the robotic simulation experiment, we store the upper bound of the score function's $\ell_2$-norm and the smallest singular value of the $H$ matrix at every time step, and obtain the following numerical results from the robotic simulation dataset:
>
> * The **smallest singular value** of the $H$ matrix is: **5.93e-1**, which clearly **satisfies** Assumption 6.
> * The upper bound of the score function's $\ell_2$-norm is: **1.51e2**.
>
> ---
>
> [1] A multi-objective/multi-task learning framework induced by pareto stationarity. ICML, 2022.
>
> [2] Enabling pareto-stationarity exploration in multi-objective reinforcement learning: A multi-objective weighted-chebyshev actor-critic approach. ArXiv, 2507.21397.

---

> ### Author Response · Authors · 2025-11-24
> **Author Response - Part 6**
>
> > **Comment 6:** The control cost is defined (only in the appendix) as a squared norm, but in Figure 3 it assumes also negative values. Also, in seems to be maximized rather than minimized (as stated in the text). Can you check this and provide the definitions completely and already when needed?
>
> **Response:** Thanks for your keen observations. The original reward design descriptions are indeed somewhat unclear. We would like to further clarify our reward signal designs as follows. In the robotic simulation task, the two reward signals designed for the objectives are defined as follows:
>
> * `Move = Health + Forward`: This objective encourages the robot to move forward as quickly as possible while preventing it from falling.
> * `Control = Health - Cost`: This objective aims to minimize the control cost while also ensuring that the robot remains standing.
>
> To avoid similar confusions in the future, the above detailed definitions have been added in Lines 1225-1233 of Appendix C.2 in our revision, and all involved figures have also been updated. Based on these reward setup, we confirm that **(i)** the values of the "Control" objective can be either positive or negative, and **(ii)** for both objectives, the robot aims to **maximize** the set of multiple objective functions.

---

### Author Response · Authors · 2025-12-03
**General Response - Part 3**

## III. Other Clarifications


Besides Assumption 4, we also clarified several technical assumptions as follows:

* **Assumption 6 Is Moderate and Nonrestrictive:** To address concerns raised by several reviewers, we have provided a more intuitive explanation for Assumption 6. Specifically, Assumption 6 only requires the matrix $H(\theta|p)$, which is "tall and skinny", to have full column rank, a condition that is typically satisfied in practice. We have also provided numerical results to further validate Assumption 6.
* **Guidance on Parameter Selection:** We also clarified that, in addition to offering a theoretical finite-time convergence rate, Theorem 1 (our main result) also provides practical guidelines on how to choose the key parameters in our proposed algorithm. Specifically, the selection of the DNN size, the smoothed coefficient, and the batch sizes can all be guided by Theorem 1.
* **Additional Related Works Survey:** As suggested by Reviewer bj58, we have added discussions on more related works in the MORL literature.

We sincerely thank all reviewers' insightful observations, suggestions, and questions, which have helped us significantly improve our paper.


## IV. Further Numerical Experiments

During the rebuttal period, we followed several reviewers' suggestions to enhance our numerical experiments:

* **Clarifying Experimental Settings:** Following the suggestions from Reviewers ePuK, LnkA, and E4KY, we have further explained the experimental settings used in our two experiments. Specifically, (i) the preference vectors used in the recommendation system experiment are not one-hot vectors; instead, we selected various $p\in\Delta_m^{++}$ to systematically explore the entire Pareto front; and (ii) we have confirmed the reward design in the robotic simulation experiments. We have incorporated these changes in the revised version of the paper.
* **Other Performance Metrics:** As suggested by Reviewers LnkA and E4KY, we have evaluated our proposed algorithm using the HyperVolume and $\epsilon$-metric, both of which are widely adopted in the multi-objective literature. The newly added results with these two performance metrics consistently demonstrate the superior performances of our method over the baselines.
* **Using More Preference Vectors to Characterize the Pareto Front:** After checking the references suggested by Reviewer E4KY, we have confirmed that our preference selection method based on (nearly) uniformly grid points in the standard simplex is actually **standard** and **widely used** in the literature. To further enhance our grid-point-based method for characterizing the Pareto front, during this rebuttal period, we have incorporated randomly preference vector sampling. Specifically, we sample additional preference vectors uniformly at random to fill the "sparse" regions on the discretized representation of the Pareto front obtained based on the grid-point approach. Our results show that this enhanced approach has a much more even coverage on the Pareto front.
* **Statistical Information and Ablation Studies:** Following the suggestions of Reviewer ePuK, we further improved our experiments as follows: (i) we have added 95% confidence intervals in Fig. 5 in the revision to enrich our numerical results; and (ii) we have conducted ablation studies to clearly demonstrate that our smoothed-WC-based approach outperforms the linear scalarization-based approach.

We sincerely thank the reviewers for their suggestions on our numerical experiments. We believe that these additional experimental results following the reviewers' suggestions have significantly improved the paper.

---

In summary, during the rebuttal period, we carefully addressed all of the reviewers' concerns, including but not limited to highlighting our contributions, fixing technical issues, adding extensive numerical experiments, and updating revisions to enhance the quality of our paper. Although we are no longer able to further engage with the reviewers, we hope our responses have satisfactorily addressed most of their comments, questions, and concerns. Thanks also go to all the ACs who have taken their precious time to review our summary and responses, as well as evaluate our paper.

Best regards,\
Authors of Paper 14511

---

### Author Response · Authors · 2025-12-03
**General Response - Part 2**

## II. Reconciling the Conflict Between Assumption 4 and Lemma 2 and Relaxing Assumption 4

We sincerely thank Reviewer LnkA for pointing out the conflict between Assumption 4 and our use of ReLU-based MLPs in the original Lemma 2, which arises from ReLU’s "scale-invariant" property. We acknowledge that this was indeed an oversight. We have since shown that this conflict can be resolved by replacing the activation function with Sigmoid, which avoids the scale-invariance issue while retaining comparable universal approximation capability. This fully addresses Reviewer LnkA’s primary concerns (cf. Reviewer LnkA's acknowledgement).


Regarding Reviewer LnkA's further questions on the potential existence of null vectors in MLPs (due to MLPs' non-injectivity) that could still violate the Fisher non-degeneracy (FND) condition in Assumption 4, we have also provided a detailed response as follows:

* **Relaxing Assumption 4 by Slightly Modifying Assumption 5:** We agree with Reviewer LnkA that the injective property does not hold for MLPs in general and null vectors could exist in MLPs. However, besides the anecdotal cases (e.g., "dead neurons" suggested by Reviewer LnkA), the fundamental reason of the existence of such null vectors arises from potential "dimension reduction" between two consecutive layers in an MLP. Specifically, if there is any "short and fat" weight matrix between two consecutive layers in an MLP (i.e., projecting from a higher dimensional latent space at one layer to a lower dimensional latent space at the next layer), a non-empty null space will exist.

  To resolve this fundamental issue, our solution is to **relax** the FND requirement in Assumption 4 and slightly modify Assumption 5 by imposing a small perturbation requirement (cf. our new Assumption 5). Specifically, we note that the main purpose of assuming FND in Assumption 4 is to ensure that the norm of Fisher matrix $\\|F(\theta)\\|$ can be lower bounded away from zero by some $\sigma>0$, which facilitates a necessary step in many related works in the literature to achieve global optimality (see, e.g., Sec. 11 in [2]). Now, with the newly added $\sigma$-perturbation in Assumption 5, we can directly guarantee the $\\|F(\theta)^{-1}\\| \le \frac{1}{\sigma}$ instead. By using the modified Assumption 5, our proofs and results will continue to hold, while avoiding the use of FND in Assumption 4 and any potential issues caused by MLPs' non-injectivity.

* **FND in Assumption 4 Holds for MLPs with Non-decreasing Widths:** Although FND in the original Assumption 4 does not hold for MLPs in general, it is worth noting that FND can be nonetheless guaranteed by MLPs with non-decreasing widths (i.e., the width of one layer is never wider than its next layer). Also, any weight matrix between two consecutive layers should be of full column rank. Under these structural requirements, we can guarantee the Sigmoid-based MLPs do not have an empty null space, which further implies FND in Assumption 4. We also note that the above "non-decreasing width" and "full column-rank" MLP structural requirements can be viewed as a natural **extension** of the "full column-rank" condition for linear policies (e.g., [3]) to **nonlinear neural MLP policies**.

---

[2] On the Global Optimum Convergence of Momentum-based Policy Gradient. AISTATS, 2022.

[3] An Improved Analysis of (Variance-Reduced) Policy Gradient and Natural Policy Gradient Methods. NeurIPS, 2020.

(See **General Response - Part 3**)

---

### Author Response · Authors · 2025-12-03
**General Response - Part 1**

Dear Area Chairs:

We sincerely thank you and all the reviewers for their time, effort, and the constructive, insightful feedback on our work. We have made every effort to address all concerns within the rebuttal period. However, since further feedback was no longer possible due to the shortened discussion period, we summarize below the key points from our rebuttals that directly address the reviewers’ comments. We hope this will assist the ACs in forming a clear and fair evaluation of our paper.

## I. Introduction of Our Work and Summary of Key Contributions

In this work, we consider the **open problem** of achieving systematic **Pareto front exploration** for **deep-neural actor-critic multi-objective reinforcement learning** (MORL) with **theoretical finite-time global convergence rate guarantee**. However, this problem is highly non-trivial due to the following technical challenges:

1. The conventional weighted Chebyshev (WC)-scalarization technique known for systematic Pareto front exploration in multi-objective optimization (MOO) is **fundamentally unfit** for actor-critic policy gradient algorithm design due to its **non-smooth min-max** nature.
2. Theoretical convergence and sample complexity analysis for deep-neural-network (DNN)-based actor-critic reinforcement learning with **nonlinear function approximation** are far more challenging and complex compared to actor-critic MORL methods based on **linear** function approximations, which necessitates new proof techniques.
3. Most existing theoretical works on reinforcement learning in the literature are limited to stationary convergence guarantee, rather than global convergence guarantee. Notably, global optimality in the Pareto sense is **fundamentally different** from that in scalar-valued problems.

Our work is centered around overcoming the above technical challenges. Our key contributions are summarized as follows:

**1) A New Smoothed WC-Scalarization Approach for Pareto Front Exploration with DNN-Based Actor-Critic MORL:** To address the non-smoothness of the standard WC-scalarization, we propose a new smoothed-weighted Chebyshev (WC) scalarization technique to enable the use of policy gradients with WC-scalarization, while preserving the salient Pareto front exploration capability of the original WC-scalarization.

**2) A Judiciously Designed DNN-Based Actor-Critic Algorithms:** Through the bridge of our proposed smoothed WC-scalarization, we develop a DNN-based actor-critic algorithmic framework with a **new** $Q$-function-based actor-critic design that significantly lowers the computational costs.

**3) New Finite-Time Global Pareto-Optimality Convergence Rate Results for DNN-Based MORL:** With the techniques and insights from the previous two contributions, we further show that our proposed DNN-based actor-critic algorithm is the **first MORL method** that enjoys **global Pareto-optimality convergence** with an **$\mathcal{O}(1/T)$ finite-time convergence rate**, where $T$ denotes the total number of iterations. Moreover, our method guarantees systematic exploration of the entire Pareto front.

**4) Our Theoretical Analysis Provides New Insights of Independent Interest:** In our theoretical analysis, we demonstrate (i) how performance difference lemma (Lemma 6.1 in [1]) can be made compatible with the MORL framework, (ii) why our algorithmic design effectively bridges the sub-optimality gap by computing a **new update direction** $d$, and (iii) the impacts of DNNs' width and length in the analysis and how to analyze them. These new ideas and techniques used in our work could be of independent interest to the general fields of multi-objective optimization (MOO) and reinforcement learning (RL).

---

[1] Approximately optimal approximate reinforcement learning. ICML, 2002.

(See **General Response - Part 2**)

---

### Meta-Review · Area_Chair_ACxe · 2026-01-06

**Summary:**

This work proposes a deep neural network-based multi-objective reinforcement learning (MORL) approach and gives theoretical convergence guarantees. The proposed method is based on scalarizing the original vector-based MORL problem using the weighted Chebyshev (WC) technique and a smoothing strategy. The authors claimed that this is the first work to establish theoretical convergence guarantees for deep neural actor-critic MORL and included numerical results.

The paper received mixed reviews with scores 6, 2, 6, 4. In particular, reviewers with lower scores raised critical concerns regarding the core theory assumptions, novelty, and experimental design of the paper. After going over the paper as well as all the author-reviewer discussions by myself, I agree with the reviewers that the initial version of the paper is flawed, as also acknowledged by the authors. While the authors have provided substantial additional material during the rebuttal, I think such amount of revision would warrant another round of review and hence recommend rejection.

**Reviewer Concerns:**

Notably, Reviewer LnkA argued that the paper's initial global convergence claim is undermined because Assumption 4 on Fisher non-degeneracy cannot hold for ReLU MLP policies. The authors agreed this was an oversight and proposed replacing ReLU with Sigmoid or other non-scale-invariant activations to remove the specific null-space construction by the reviewer, and further proposed to address the more general non-injectivity concern by relaxing the original assumptions. However, the latter fix was not acknowledged by the reviewer, leaving it unclear whether the issue was fully resolved.

Reviewers ePuK and E4KY noted that the WC scalarization itself is not novel, posing a concern about the theoretical contribution. The authors have clarified this point in the rebuttal, yet not convincingly, according to the response of Reviewer E4KY.

Finally, Reviewers LnkA, E4KY, and ePuK raised concerns about the experiments, including unclear preference-vector choices, fairness of baseline comparisons, insufficient PF quality metrics, etc. The authors responded by clarifying some points and conducting additional experiments and ablations, yet some concerns remain for Reviewer LnkA.

**Reviewer Scores:**

Reviewer LnkA and E4KY (with scores 2 and 4) participated in the discussion and intended not to change the score. Other reviewers did not participate, and I feel there is little chance for them to raise the score.

---

### Decision · Program_Chairs · 2026-01-26

Reject